# Calibration of parameters in microscopic traffic flow simulation models considering micro-meteorological information

Jian Ma[1], Yuchen Zhang[1], Liyan Zhang [1]*, Zongwei Gao[1], Keyi Cao[2], Qianlong Fu[1], Zheng Qian[2]

**1** School of Civil Engineering, Suzhou University of Science and Technology, Suzhou, China, **2** School of Business, Suzhou University of Science and Technology, Suzhou, China

* outerspace@mail.usts.edu.cn

## Abstract

Different micro-meteorological conditions can affect a driver's judgment of road conditions, leading to changes in following behavior. On rainy days, water films on the road reduce traction, increasing the likelihood of hydroplaning and traffic accidents. While there are existing following models under various weather conditions, research on the specific impact of micro-meteorological factors is insufficient. To achieve fine management in intelligent transportation and real-time monitoring of vehicle states, it's essential to study following behavior under different micro-meteorological conditions and establish corresponding models. This paper focuses on the Intelligent Driver Model (IDM) and the Wiedemann99 model, considering the impact of micro-meteorological conditions. By incorporating a driver's judgment factor, λ, the IDM and Wiedemann99 models are improved, leading to the development of new models: I-IDM and I-Wiedemann99. Simulation validation is used to choose speed and following distance as performance indicators for parameter calibration of the I-IDM and I-Wiedemann99 models, with the sum of Root Mean Square Percentage Error (RMSPE) as the goodness-of-fit function. Comparisons are made between the driving paths, speeds, and accelerations of following vehicles before and after calibration, verified through simulations. The conclusions are as follows: the average error and standard deviation of the improved I-IDM model are smaller than those of the I-Wiedemann99 model, with the maximum Root Mean Square Percentage Error (RMSPE) for I-IDM model parameter calibration being 0.4568 and the minimum being 0.1324. For the I-Wiedemann99 model, the maximum RMSPE is 0.4613 and the minimum is 0.1376. The parameter calibration results of the I-Wiedemann99 model are more dispersed compared to those of the I-IDM model, indicating that the I-IDM model simulates following behavior more effectively than the I-Wiedemann99 model. The findings of this study can provide a reference for further improving the theory

**Data availability statement:** All relevant data are within the paper and its Supporting Information files.

**Funding:** This study was supported by the following sources: Collaborative Education Project of the Ministry of Education, (220904757090019 and 220904757094735), awarded to J.M. and L.Z.; Construction System Project of Jiangsu Provincial, (2020ZD14), awarded to J.M.; Philosophy and Social Science Projects of Universities in Jiangsu Province, (2023STYB1420), awarded to J.M.; High Level Innovation and Entrepreneurial Research Team Program in Jiangsu, (SJCX20_1117, SJCX21_1420, and KYCX21_2999), awarded to J.M.

**Competing interests:** NO authors have competing interests.

of following behavior, and offer a theoretical basis and IoT technology support for refined traffic management under rainy conditions.

---

# 1 Introduction

## 1.1 Background

Micrometeorological information has a great impact on roads, vehicles and people in the road traffic system. Under the influence of complex micrometeorological information, road characteristics, vehicle driving state and driver's physiology and psychology will have great changes, which makes road traffic safety, stability and efficiency decrease sharply, causing traffic accidents and seriously hinders China's economic and social development. However, in the traditional microsimulation model, the meteorological conditions are often ignored. Meteorological conditions are important factors in traffic behavior, such as rain intensity, temperature, humidity, wind speed and visibility, which will affect drivers' behavior and road conditions. Therefore, it is necessary to take micrometeorological information into consideration in the traffic simulation model. The influence of micrometeorological information on traffic flow can be considered by adding parameters of micrometeorological factors to the simulation model.

Therefore, this paper takes micro-meteorological information into micro-traffic flow simulation model to better study traffic problems, help us better understand the impact of meteorological conditions on traffic behavior, and explore micro-traffic flow simulation model considering micro-meteorological information, so as to improve the efficiency and accuracy of traffic planning and management.

In order to meet the needs of the rapid development of urban roads and the traffic under various all-weather rainfall environments, this paper studies the parameters of the micro-traffic flow simulation model under different rainfall intensification. The improved following model takes into account the drivers' judgment factors on road conditions under micrometeorological conditions, and carries out parameter calibration and simulation evaluation to verify the effectiveness of the improved model. Its main significance is as follows:

On a theoretical level, this study aims to explore the need for safety and efficiency in smart cities under rainfall conditions. Through in-depth analysis, it reveals rainfall affects the micro-traffic flow parameters of urban roads, and elucidates the reasons for macro-phenomena such as the slowing down of urban road traffic flow speed and the reduction of traffic capacity in rainy weather. Therefore, this paper will build a rainy day follow through model to enrich the theoretical basis of micro follow through behavior. At the application level, this paper will calibrate and verify the following model under different rainfall intensity, so as to apply it to many fields such as micro-traffic simulation, traffic capacity analysis, traffic management and traffic safety evaluation. In addition, the accurate calibration of the parameters of the following model will help promote the improvement of traffic safety and transportation efficiency. Finally, the combination of micro-traffic flow simulation model and

micro-meteoro logical information and the application of visualization technology is a key direction of traffic flow research, which helps us to understand the relationship between traffic behavior and meteorological conditions more deeply, so as to improve the efficiency and accuracy of traffic management and planning.

## 1.2 Literature review

Numerous research results have been achieved in the study of the impact of rainfall on traffic flow characteristics. These results generally show that rainfall will lead to the decrease of traffic flow speed and capacity on urban roads, and also change traffic flow parameters such as expected speed and time headway. This means that although there is a certain understanding of the influence of rainfall on the macro characteristics of traffic flow, there is still a need for further research and exploration on how rainfall affects individual driving behavior and how to describe these micro behaviors through models.

Since the early 1950s, Reuschel and Pipes have conducted dynamic analysis of traffic flow from the perspective of vehicle dynamics, and since then, the prototype of stimulus-response model has begun to appear [1]. During the late 1950s and early 1960s, many scholars conducted a lot of research based on the stimulus-response model of Pipes, which promoted the development of the stimulus-response model. In 1959, Kometani et al. proposed the first safe distance class model [2]. In 1961, Gazis et al. summarized previous research results based on several recognized basic assumptions in following behavior, and proposed a classical GHR (Gazis-Herman-Rothery) model, which greatly promoted the development of traffic flow theory [3]. In 1963, Michaels proposed the first psychophysiological model, which was established based on the visual psychological hypothesis, and believed that the driver could judge whether the lead vehicle was approaching or away from the other vehicle by perceiving the changes in the rear area of the lead vehicle in the field of vision [4]. In 1981, Gipps et al. developed a Gipps model based on the requirement of software simulation to reproduce real traffic flow characteristics, which was used in the traffic simulation software AIMSUN [5].

In 1973, Evans et al. carried out several simulation experiments and re-calibrated the threshold based on Michaels model [6]. In 1974, Wiedemann further defined the driver's perception and response threshold, which became the core module of the famous simulation software VISSIM [7]. Since then, human factors have been paid more and more attention in the modeling process of the following model, and important factors such as the driver's expected speed, expected distance and expected acceleration have been discovered by scholars and introduced into the following model. In 1995, Bando et al. proposed the Optimal Velocity (OV) model, which was widely used to simulate macroscopic traffic flow phenomena [8]. In 1998, Helbing et al. revised the OV model and proposed the General Force (GF) model, which shows the influence of the acceleration of the GF model on the condition that the speed of the vehicle in front is less than that of the vehicle behind [9]. In 2000, Jiang Rui et al. further modified the GF model of Helbing and proposed the Full Velocity Difference (FVD) model by more comprehensively considering the influence of positive and negative velocity difference between the front and following vehicles during the following process [10]. In 2000, Treiber et al. proposed an Intelligent Driver Model (IDM) that considered the driver's desired vehicle distance and desired speed, and analyzed and learned a large amount of following data through machine learning algorithms, without relying on traditional following theory as the basis for modeling [11]. Therefore, with the progress of traffic flow data acquisition technology, the following model based on artificial intelligence has been further developed and improved. In 2012, Tang et al. considered the influence of car-following behavior under different road conditions and established a car-following model under the influence of road conditions based on the optimized speed class model [12]. In 2018, Zhu et al. simulated the traffic flow when the human-autonomous vehicles mixed, and proposed a car-following model suitable for the autonomous vehicles based on the optimized speed model [13]. In 2020, Yang Longhai et al. [14] studied and analyzed the driving behavior changes of drivers in different traffic states under ice and snow weather, and established a car-following model under ice and snow conditions based on IDM model.

In the field of car-following models, numerous research achievements have been made. However, further research and exploration are needed to understand how rainfall affects individual car-following behavior and how to describe these microscopic behaviors through models.

Heavy rainfall and other adverse weather conditions significantly impact traffic flow, affecting macro-level characteristics such as traffic speed and flow, as well as micro-level behaviors like vehicle-following and lane-changing [15]. Car-following models are essential for explaining many microscopic traffic phenomena and reflecting the nature of traffic flow. Although extensive research exists on establishing these models [16], fewer studies focus on their calibration. For example, Arne et al. used vehicle trajectory data and genetic algorithms to optimize model parameters [17]. Wang Dianhai and colleagues calibrated the GM model using measured data [18], while Jiang Jun's team used real vehicle experiments to calibrate the Helly model [19]. Classic models like GM, NEWELL, Helly, Gipps, OV, and FVD have been developed under normal weather conditions [10], failing to account for adverse weather factors like rainfall. Consequently, their calibrated parameters do not accurately reflect traffic flow characteristics during such conditions. Research on car-following models and their parameter calibration under the influence of rainfall is limited. For instance, Gong Jiekun introduced adhesion and slip rate parameters in a model considering real-time road conditions [20], while Yang et al. developed a model that accounts for maximum deceleration, comparing its performance on dry and icy surfaces [21].

In summary, current research on car-following models has become more comprehensive in its considerations. With the focus primarily on the impacts of external factors (adverse weather conditions) and internal factors (transportation infrastructure, vehicle characteristics, driver behavior), as well as heterogeneous traffic flow (mixed motorized and non-motorized traffic, mixed human-driven and autonomous vehicles) on car-following behavior. By refining existing models, scholars have been able to develop car-following models that account for the influence of these stochastic factors.

Based on previous research and practical application requirements, this paper hopes to select a suitable micro traffic flow simulation model for simulation experiments, apply the determined parameters to the simulation model, conduct simulation experiments combined with micrometeorological information, and record the simulation results. According to the simulation results and the objective function, the parameter calibration algorithm is used to calibrate the parameters of the simulation model to improve the accuracy and reliability of the model.

## 2 The establishment of micro-traffic flow simulation model considering micro-meteorological information

Considering the limitations and applicability of various following models, this paper selected IDM model with clear and mature model parameters, and Wiedemann99, the main following model in Sumo and PTV Vissim, which can describe the driver's following behavior in a more detailed way under different rainfall intensities, as the model studied in this paper.

### 2.1 Analysis of traffic flow mechanism characteristics under adverse weather

The purpose of analyzing the mechanism characteristics of traffic flow under adverse weather conditions is to understand the operation rules and characteristics of traffic flow under adverse weather conditions, so as to take effective traffic management measures to maintain road traffic safety and smooth passage. The analysis of traffic flow mechanism characteristics under adverse weather conditions is conducive to improving the scientific nature of traffic management and improving road traffic safety.

The characteristics of traffic flow mechanism under adverse weather conditions include reduced visibility: in severe weather conditions such as rain and snow, vehicle visibility is reduced, and the driver's vision is worse, which may lead to traffic accidents. In addition, reduced visibility will also affect the driver's reaction time and driving speed, and then affect the road capacity and traffic congestion. Wet road slippery: In weather conditions such as rain and snow, the road will become wet, the vehicle braking distance increases, and it is easy to slip or lose control. Slippery roads may also lead to a decrease in vehicle speed, which affects road capacity and traffic congestion. Increased vehicle spacing: In weather conditions such as rain and snow, vehicle spacing increases and drivers shorten their driving speed to maintain a safe distance. Increased vehicle spacing can lead to reduced road capacity, which can lead to traffic congestion. Increased traffic

accidents: The incidence of traffic accidents increases during adverse weather conditions. The occurrence of accidents may cause problems such as road closures, traffic congestion, etc., which can affect road capacity.

Ideal weather should have the following conditions:

(1) No rainfall;

(2) Dry ground;

(3) Good visibility or visibility greater than 4 km;

(4) Wind speed not greater than 3m/s.

The classification standards for rainfall intensity by the meteorological department are shown in Table 1.

As long as one of the above conditions is not met, it is unfavorable weather. Because snow can be converted into rain, it can be divided into two types according to whether there is precipitation or not. This paper chooses rainy day as the research background. The influence of different rainfall intensity on traffic flow is different, and the traffic flow data under different rainfall intensity are studied respectively. The classification criteria of rainfall intensity of China's meteorological department are shown in Table 1.

**2.1.1 Analysis of the influence of different rainfall intensity on following behavior.** The road traffic system is a complex system composed of multiple factors. The driver is the core of the whole system, whose driving behavior directly affects the motion state of the vehicle and the road capacity. The driver needs to constantly perceive information, such as road conditions, traffic conditions, weather environment, etc., while information processing is needed to make appropriate responses according to different road and traffic conditions, so that the vehicle can travel as expected. Such responses include actions such as acceleration, deceleration and steering, as well as following and overtaking of other vehicles.

In rain conditions, the following behavior changes. As the friction coefficient of the road decreases and the braking distance of the vehicle increases, the driver needs to increase the following distance to ensure driving safety. At the same time, rain will also affect the speed of vehicles, drivers may take a more conservative driving strategy, slow down the speed to maintain a safe distance. In addition, the driver's expectation of driving will also change under the condition of rain, and it is necessary to make corresponding adjustments to the rainy driving environment to cope with emergencies. Therefore, the mechanism of the influence of rainfall on human-vehicle-road is very complicated, and it needs to be deeply analyzed and studied, so as to provide theoretical support for formulating more scientific and reasonable traffic management measures.

**2.1.2 Analysis of the impact of rainfall on vehicles.** The impact of rainfall on vehicles is mainly reflected in the lower friction coefficient between road surface and tire in rainy weather, which affects the braking performance of vehicles. Under the condition of rainfall, as shown in Fig 1, the road is covered with a layer of water film, which will reduce the adhesion between the vehicle tire and the road surface. When the water that lands on the road cannot be discharged in time, resulting in the water film reaching a certain thickness, the vehicle will even have the phenomenon of rowing. This phenomenon is due to the appearance of water film leads to the vehicle tire and the road in the gap, so that the tire completely off the road, then the vehicle brake and direction control may fail, this phenomenon will cause a great safety

**Table 1. Classification criteria of rainfall intensity of meteorological department.**

| Rainfall Types | Rainfall (mm) | | |
|---|---|---|---|
| | Rainfall in 1h | Rainfall in 12h | Rainfall in 24h |
| Light Rain | rain_IY ≤ 2.5 | rain_IY<5 | rain_IY<10 |
| Moderate Rain | 2.6 ≤ rain_IY ≤ 8.0 | 5 ≤ rain_IY ≤ 14.9 | 10 ≤ rain_IY ≤ 24.9 |
| Heavy Rain | 8.1 ≤ rain_IY ≤ 15.9 | 15 ≤ rain_IY ≤ 29.9 | 25 ≤ rain_IY ≤ 49.9 |

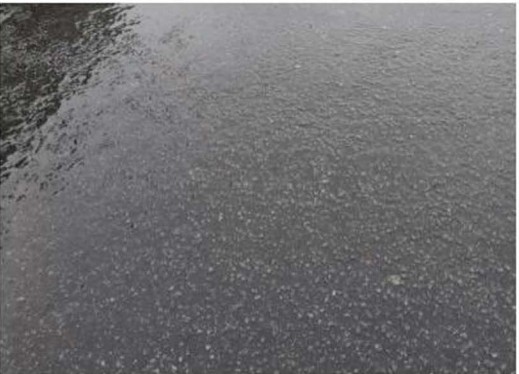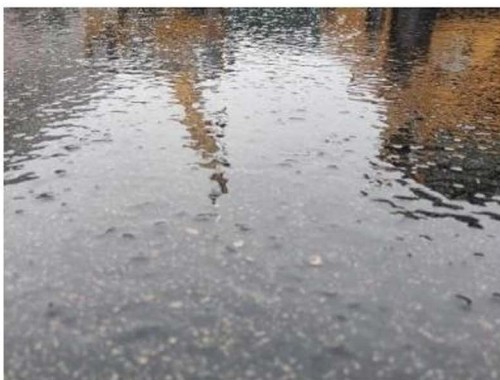

**Fig 1. Light rain(a)and heavy rain(b) film thickness under meteorological conditions.**

hazard to the driver's driving behavior. Mounce John M and Bartoskewitz T from the United States obtained the critical water-skiing formula when the thickness of the water film does not exceed 2.5mm through the relationship between the rainfall in rainy days and the speed of the vehicle:

$$v = 96.84h^{-0.259} \tag{1}$$

In the equation: $h \leq 2.5mm$.

The tire hydroplane speed of the tire when the water film thickness does not exceed 2.5 is shown in Table 2.

Ji Tianjian et al. carried out finite element analysis on the tires of the vehicle, calculated the adhesion coefficient of the road surface when the vehicle is at different speeds and the thickness of the road water film, calculated the relationship between the speed and the thickness of the water film when water skiing [22]:

$$h = 0.001v^2 - 0.2798v + 21.499 \tag{2}$$

The relationship between the vehicle speed at which hydroplaning occurs and the thickness of the water film is shown in Table 3.

The comprehensive analysis shows that when driving in rain, the braking performance of the vehicle will be significantly affected, resulting in an increase in the time and distance required for braking. In order to ensure driving safety and improve the handling ability of the vehicle, drivers usually choose to increase the distance from the surrounding vehicles

**Table 2. Tire hydroplaning threshold speed at water film thickness ≤ 2.5 mm.**

| Items | Units | Corresponding values | | | | | | | | | |
|---|---|---|---|---|---|---|---|---|---|---|---|
| **Criticl speed** | km/h | 138 | 116 | 104 | 97 | 91 | 87 | 84 | 81 | 79 | 76 |
| **Water Film Thickness** | mm | 0.25 | 0.5 | 0.75 | 1 | 1.25 | 1.5 | 1.75 | 2 | 2.25 | 2.5 |

**Table 3. Vehicle speed during hydroplaning vs. water film thickness.**

| Items | Units | Corresponding values | | | | | | | |
|---|---|---|---|---|---|---|---|---|---|
| **Water Film Thickness** | mm | 6.8 | 5.5 | 4.4 | 3.5 | 2.8 | 2.3 | 2 | 1.9 |
| **Water Film Thickness** | km/h | 70 | 80 | 90 | 100 | 110 | 120 | 130 | 140 |

and reduce the driving speed. In addition, the accumulation of rainwater can form a water film on the road surface,and the thicker the film, the higher the risk that the vehicle will lose control at the critical speed of water skiing. This poses a challenge for drivers to react quickly in an emergency, take braking or adjust their driving strategy, thus potentially threatening driving safety.

**2.1.3 Analysis of the impact of rainfall on road conditions.** In rainy weather, road adhesion coefficient is mainly related to driving speed and water film thickness. Slippery road surfaces increase braking distance and risk of loss of control. When the vehicle braking on a wet road surface, the braking distance will be longer than on a dry road surface, and the stability of the vehicle will be reduced in this case, and it is prone to skidding or sides carding. Therefore, the driver should reduce the speed of the vehicle, increase the following distance, and avoid sharp turns or braking. The vehicle is the carrier of the road, and the road conditions restrict the operation of the traffic. In the rainy environment, the road will become wet, if the road water cannot be discharged in time, puddles will further reduce the anti-skid performance of the road. In addition, in rainy weather, long-term water will lead to pavement damage, and the damaged road will affect the safety of vehicles and pedestrians, road surface damage and damage will be more likely to cause vehicles and pedestrians to slip and fall, and even damage to the vehicle suspension, tires and braking system, at this time, Even if the driver's defect value is low, he cannot make timely and correct driving decisions when the vehicle road environment changes.

## 2.2 General micro traffic flow simulation model

**2.2.1 IDM model.** The IDM Model is a basic model in traffic flow theory, also known as the Intelligent Driver Model, which describes the behavior of vehicles on the road. Originally proposed in 2000 by German scientists Treiber, Hennecke and Helbing, the model expresses the relationship between a vehicle's acceleration, speed and the distance between the front of the vehicle as a system of differential equations. This differential equation includes many factors, such as the ideal speed of the vehicle, the distance between the vehicle and the vehicle in front, the speed difference, braking force, etc., to simulate the driving situation of the vehicle on the road. The main feature of the IDM model is to connect the acceleration of the vehicle with the distance and speed difference between the lead vehicles, so as to realize the coordination between the vehicles. In practical applications, the model can be used in traffic flow optimization, traffic congestion control, vehicle safety control and other aspects.

As shown in Fig 2, the advantages of IDM model are that it can describe the behavior and interaction of the system intuitively, consider the driver's expected speed and expected distance, and have the advantages of few parameters, clear physical meaning, easy calibration and so on. The model can describe all the scenes of the vehicle from driving freely to following the lead vehicle and then to stop with the lead vehicle (or stop at the red light) in order to better understand the working principle and behavior characteristics of the system, flexibly change the parameters and rules for simulation, in order to evaluate the effects of various schemes and strategies, provide highly detailed analysis, in order to better understand the characteristics of the system.

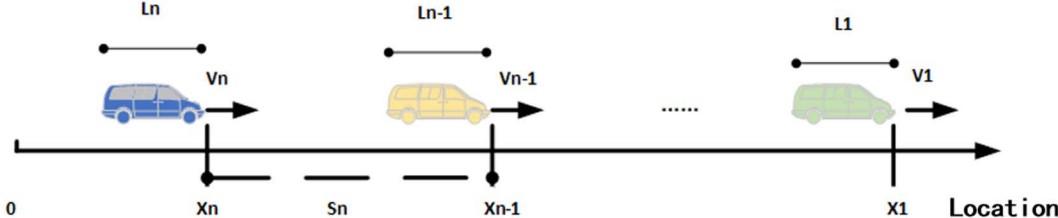

**Fig 2. Schematic diagram of IDM model following motion.**

The IDM model proposed by Treiber is based on the distance and speed between the lead and following vehicles [11]. The formula is as follows:

$$\alpha_n(t) = \alpha_{max}^{(n)} \left[ 1 - \left( \frac{V_n(t)}{\widetilde{V}_n(t)} \right)^{\beta} - \left( \frac{\widetilde{S}_n(t)}{S_n(t)} \right)^2 \right]$$

(3)

$$\widetilde{S}_n(t) = S_{jam}^{(n)} + V_n(t)\widetilde{T}_n(t) + \frac{V_n(t)\Delta V_n(t)}{2\sqrt{\alpha_{max}^{(n)}\alpha_{comf}^{(n)}}}$$

(4)

Where:

$$\begin{cases} S_n(t) = x_{n-1}(t) - x_n(t) - l \\ \Delta v_n(t) = v_n(t) - v_{n-1}(t) \end{cases}$$

(5)

Where $\alpha_{max}^{(n)}$ is the maximum acceleration or deceleration of the following vehicle; $\beta$ is the acceleration index; $\alpha_{comf}^{(n)}$ is the expected deceleration of the following vehicle; $\widetilde{v}_n$ expected speed for the driver; $S_n$ is following vehicle to lead vehicle distance; $l$ is the length of the front body $\widetilde{S}_n$ is the following distance expected by the driver; $S_{jam}^{(n)}$ is the blocking distance at rest; $\widetilde{T}_n(t)$ is the expected headway; $V_n(t)$ represents the speed of the target vehicle at the time $t$; $\Delta V_n(t)$ represents the speed difference between the target vehicle and the vehicle in front at the time $t$, $\alpha_n(t)$ represents the acceleration of the target vehicle at the time $t$. $\alpha_{comf}^{(n)}$, $\widetilde{v}_n$, $\widetilde{S}_n$, $S_{jam}^{(n)}$, $\widetilde{T}_n(t)$, are the parameters to be calibrated later.

**2.2.2 Wiedemann99 model.** Wiedemann99 driving behavior threshold model is a physiological-psychological model. The driver in the following vehicle changes the speed of the vehicle by perceiving the workshop distance and speed difference between his vehicle and the vehicle in front of him, thereby triggering the speed change mechanism of the driving behavior threshold curve. This model is applied in VISSIM and defines four different driving states [23,24].

(1) free driving: The driving behavior of the following driver is not affected by the driving behavior of the lead vehicle. In this case, the following driver will gradually reach a stable speed, which is called the desired speed.

(2) approaching: When the distance between the following vehicle and the lead vehicle becomes small, the following vehicle driver will choose to adjust the speed or brake directly until the distance between the following vehicle and the lead vehicle meets the safety distance expected by the following vehicle driver.

(3) following: there is no obvious acceleration or deceleration behavior of the following driver. The relative distance between the following vehicle and the lead vehicle fluctuates continuously at a level of safety distance.

(4) braking: When the following driver feels that the distance between the two vehicles is too small and less than the safe distance expected by the following driver, he will choose to brake, and the deceleration value of the vehicle will slowly transition to the maximum.

Wiedemann driving behavior threshold model includes two models, Wiedemann74 and Wiedemann99, which respectively show the two conditions of urban road and expressway (urban expressway). The Wiedemann99 model is selected as the research object in this paper.

The Wiedemann99 model has ten threshold parameters, one of which is the sensory threshold and the other is the driving behavior threshold.

1. CC0.

CC0 refers to the average expected distance between two stopped vehicles, that is, the blocking distance at rest. AX represents the expected stopping distance of stationary vehicles, which is mainly determined by the sum of the expected safety distance of the driver behind and the length of the vehicle in front; $L_{n-1}$ represents the length of the vehicle in front.

$$AX = CC0 + L_{n-1} \tag{6}$$

2. CC1.

CC1 stands for expected headway, which is the value of the headway that the driver behind shows that he expects to maintain for a constant speed; $v_{slower}$ represents the speed of the vehicle in front of and behind the following vehicle compared to the slower vehicle.

$$ABX = AX + CC1 \cdot v_{slower} \tag{7}$$

3. CC2.

CC2 represents the following variable and represents the constraint of longitudinal oscillation of the distance between the front and following vehicles. This constraint defines the maximum safe distance ABX for the driver behind to approach the vehicle in front.

$$SDX = ABX + CC2 \tag{8}$$

4. CC3.

CC3 represents a "following" state threshold that regulates the slowing behavior of the following vehicle. SDV is the absolute value of the difference in speed between the two vehicles when the driver behind starts to pay attention to the slow-moving vehicle ahead. $\Delta v$ is the relative speed of the vehicle in front and behind.

$$SDV : \Delta x = CC3 \cdot \Delta v + SDX + CC3 \cdot (-CC4) \tag{9}$$

5. CC4.

CC4 describes the "following" state between a following and lead vehicle. When the speed difference is $\leq$ CC4, the following vehicle adjusts to maintain a safe distance. A smaller CC4 increases sensitivity to the front driver's braking, allowing for quicker reactions. It's used when the following vehicle is slower than the front. CLDV indicates the following driver sees a decreasing distance to the lead vehicle and considers braking to ensure safety.

$$CLDV = -CC4 \tag{10}$$

6. CC5.

CC5 is similar to CC4 but applies when the following vehicle's speed exceeds the lead vehicle's. Their values are identical. In PTV Vissim, default values for CC4 and CC5 indicate a close following relationship. OPDV denotes the absolute speed difference estimated by the following driver when the vehicles are farther apart.

$$OPDV = -CC5 \tag{11}$$

7. CC6.

CC6 measures the following performance of a vehicle, reflecting the speed fluctuation of the following vehicle while following the front. This fluctuation is caused by changing gaps and shows how the following speed is influenced by the front distance. A CC6 value of 0 means speed changes are unrelated to distance. As CC6 increases, speed fluctuations become more pronounced with larger gaps, requiring more frequent acceleration and deceleration to maintain safety, leading to greater speed variability.

8. CC7.

CC7 represents the oscillating acceleration.

9. CC8.

   CC8 is the acceleration of stopping.

10. CC9.

CC9 represents the expected acceleration of the vehicle at 80 km/h.
   The schematic diagram of Wiedemann99 model following motion is shown in Fig 3.

## 2.3  Improved micro-traffic flow simulation model considering micro-meteorological information

Rainy weather will lead to wet road surface, change the friction coefficient between the vehicle and the road surface, increase the braking distance and the risk of vehicle loss of control, while under complex micrometeorological conditions,

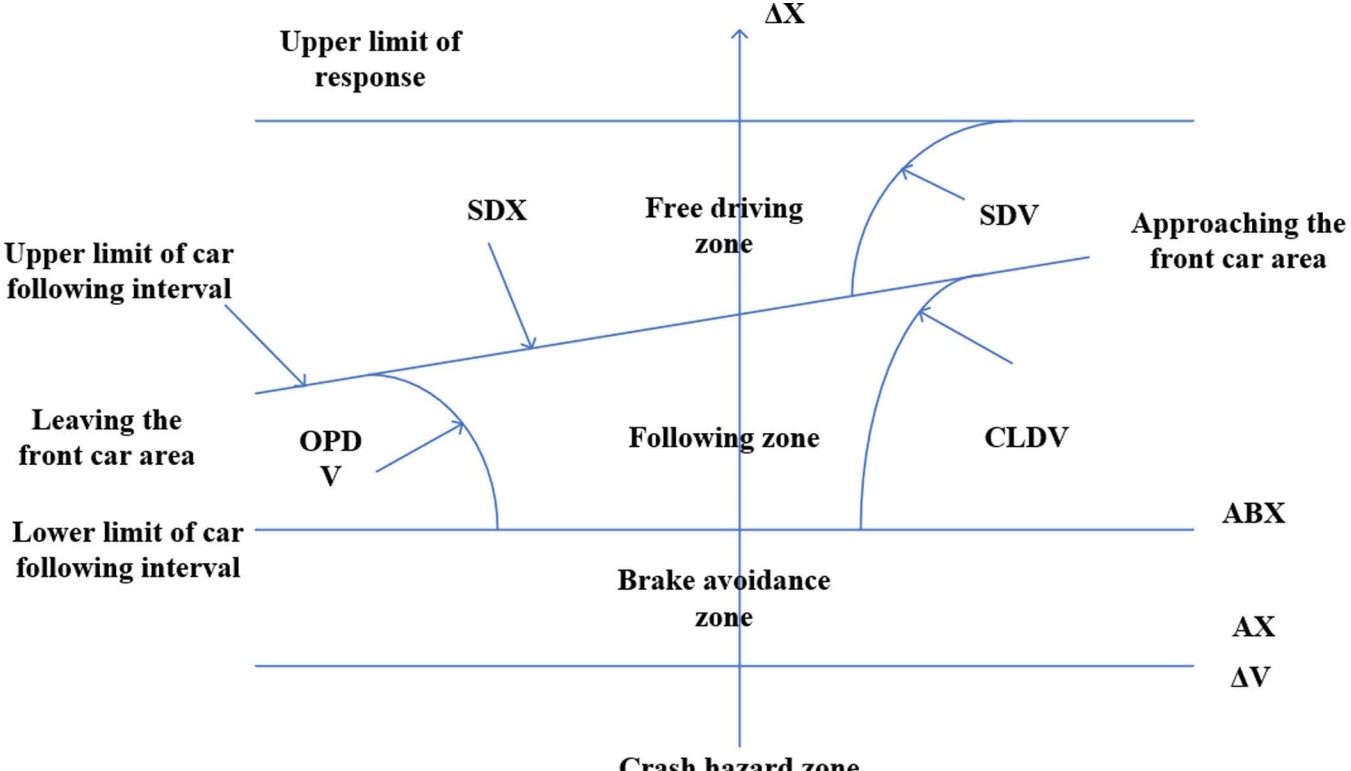

**Fig 3.  Schematic diagram of the Wiedemann99 model fol lowing motion.**

the driver needs more time to deal with and respond to changes in the external environment, which poses a challenge to the driver's reaction speed and judgment ability [25].

In order to make the car-following model better reflect the driving behavior of different drivers under different micrometeorological conditions, the parameter $\lambda$, which affects the judgment of drivers on road conditions, is introduced into the model. In order to avoid the adverse effect of too large or too small influence factors on the experiment, resulting in false judgment of the following vehicle on the speed and position of the lead vehicle, the value $\lambda \in [0.6, 0.9]$ is selected in this paper. The value of $\lambda$ is defined in this paper to be proportional to the magnitude of the rain intensity. The greater the rain intensity, the greater the influence of the environment on the driver's judgment of the road condition. The easier it is for the driver behind to conclude that he should slow down. At the same time, the larger the value is, the easier it is for the driver to keep a safer distance at a safer speed (Table 4)

(1) Improved IDM Following Model (I-IDM)

Drivers' judgment factors on road conditions $\lambda$ will have an impact on $\Delta v_n(t)$ and $S_n(t)$, from which it can be concluded that:

$$\Delta v_n(t) = v_n(t) - \lambda v_{n-1}(t) \tag{12}$$

$$S_n(t) = \lambda \left( x_{n-1}(t) - x_n(t) \right) - I \tag{13}$$

$\lambda v_{n-1}$ represents the driver's estimation of the speed of the vehicle in front, $\lambda \left( x_{n-1}(t) - x_n(t) \right)$ represents the driver's estimation of the distance between the vehicle and the vehicle in front, and the finally improved IDM model is shown as follows:

$$\alpha_n(t) = \alpha_{max}^{(n)} \left[ 1 - \left( \frac{V_n(t)}{\widetilde{V}_n(t)} \right)^{\beta} - \left( \frac{\widetilde{S}_n(t)}{\lambda \left( x_{n-1}(t) - x_n(t) \right) - I} \right)^2 \right] \tag{14}$$

$$\widetilde{S}_n(t) = S_{jam}^{(n)} + V_n(t) \widetilde{T}_n(t) + \frac{V_n(t) \left[ v_n(t) - \lambda v_{n-1}(t) \right]}{2 \sqrt{\alpha_{max}^{(n)} \alpha_{comf}^{(n)}}} \tag{15}$$

(2) Improved Wiedemann99 following model (I-Wiedemann99)

Since the Wiedemann99 model has a more detailed physiological and psychological impact on drivers, drivers' judgment factors on road conditions can be directly affected by formula (7). Considering that drivers' driving state is partly caused by personality and partly by driving environment, when the micrometeorological conditions change, the driver's judgment on road conditions will change accordingly. And the greater the rain intensity, the more cautious the driver's driving behavior. Based on this consideration, the influencing factors of expected headway CC1 can be decomposed into personality influencing factor $k$ and road condition judgment factor $\lambda$.

**Table 4. The relationship of parameters A and B to expectations and workshop spacing.**

| Parameters | Relation to Expectation and Workshop Distance |
|---|---|
| k | The larger the k, the more cautious the driver expects the greater the distance from the workshop |
| λ | The larger the λ, the more complex the change in micrometeorological conditions makes the driver judge the road condition, and the greater the expected distance from the workshop |

 

The formula for calculating the headway is as follows:

$$T = \frac{X_{n-1}(t)}{\Delta v_n(t)}$$ (16)

$X_{n-1}(t)$ is the driving distance of the vehicle in front, and the expected headway expression based on the above formula is expressed as:

$$ExpectedHeadway = \frac{kX_{n-1}(t)}{v_n(t) - \lambda v_{n-1}(t)}$$ (17)

The maximum safe distance for the driver behind to approach the lead vehicle can be improved as:

$$TheMaximumSafeDistance = AX + \frac{kX_{n-1}(t)}{v_n(t) - \lambda v_{n-1}(t)} \cdot v_{slower}$$ (18)

## 3 Parameter calibration and verification of I-IDM model and I-Wiedemann99 model

In essence, the parameter calibration of the following model is a continuous optimization process. Based on different data acquisition methods, a set of optimal model parameters can be obtained so that the model can better simulate the vehicle trajectory. The smaller the difference between the simulated vehicle trajectory and the actual vehicle trajectory, the better the calibration result. At the same time, it is necessary to combine the simulation or field data to determine the input variables and output variables of the model, so as to use mathematical methods to solve the undetermined parameters in the model. Through parameter calibration, the following model can be used to describe the following behavior and be applied in practice.

In general, as shown in Fig 4, the parameter calibration of the following model goes through five stages: ①The actual trajectory of the front and following vehicles is obtained by Sumo;②the simulation trajectory of the following vehicle is obtained by the optimization algorithm, which includes Newton descending method, genetic algorithm, particle swarm algorithm, etc. ③the trajectory and motion state of the following vehicle generated by the optimization algorithm and the

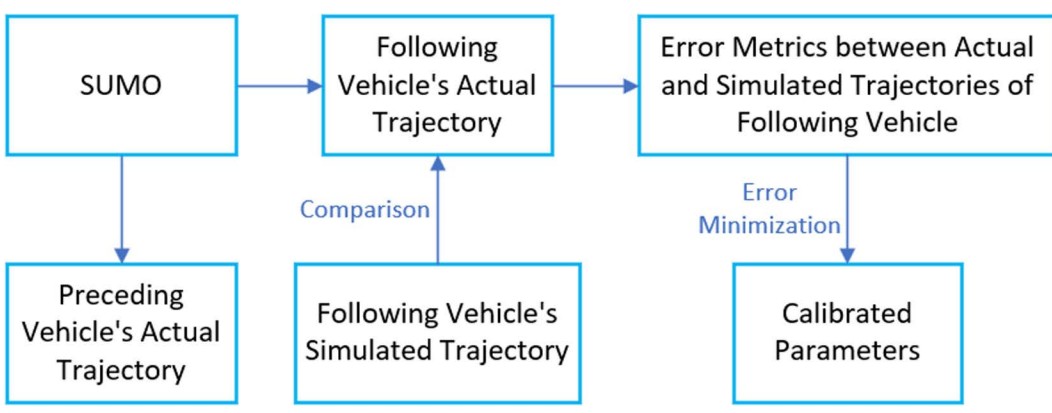

**Fig 4. Following model parameter calibration flow chart.**

trajectory and motion state of the following vehicle after Sumo simulation;④The error indexes of the real track and the simulated track of the following vehicle are obtained;⑤the parameter that minimizes the error is the result of the calibration parameter.

(1) Determine the goodness of fit function

The goodness of fit function is one of the commonly used indicators to measure how well a model fits the data. It is often used to compare the difference between a model's predictions and the actual values, and to calculate the error between them. The smaller the goodness of fit function value, the smaller the prediction error and the better the performance of the model.

In this paper, the sum of Root Mean Square Percentage Error (RMSPE)is chosen as the goodness of fit function. RMSPE is a goodness of fit function widely used in statistics and machine learning that measures the relative error between a model's predicted value and its true value [26,27]. Compared to other goodness of fit functions, RMSPE is effective at avoiding situations where the predicted value is too high or too low, and can take into account the range of variation in the data when evaluating it. A smaller RMSPE value means a smaller error between the calibrated model and the actual situation.

The formula for RMSPE is as follows:

$$E_{RMSPE} = \sqrt{\frac{\sum_{i=1}^{N} \left(s_i^{sim} - s_i^{obs}\right)^2}{\sum_{i=1}^{N} \left(s_i^{obs}\right)^2}} + \sqrt{\frac{\sum_{i=1}^{N} \left(\nu_i^{sim} - \nu_i^{obs}\right)^2}{\sum_{i=1}^{N} \left(\nu_i^{obs}\right)^2}}$$

(19)

Where: $s_i^{obs}$ is actual distance headway value of the $i$ sample; $s_i^{sim}$ represents output value of simulation model of distance headway of the $i$ sample after calibration; $v_i^{obs}$ is the actual speed value of the $i$ sample vehicle, $v_i^{sim}$ represents the output value of the $i$ sample speed simulation model after calibration; $i$ is the observation sequence and $N$ is the total number of observed samples.

(2) Optimization algorithm for solving

The calibration of the parameters of the following model needs to take the goodness of fit function as the objective function, and use the optimization algorithm to solve the parameters of the model to be calibrated. For the optimization of nonlinear problems, the commonly used algorithms include iterative method and intelligent algorithm.

After the parameter calibration of the following model is completed, the model needs to be evaluated and verified. Only the verified model can guarantee its accuracy and effectiveness. This study will evaluate and validate the model using the following two methods: (1) Analyze the errors of different performance indicators in the stages of model calibration and verification. The smaller the errors, the better the performance of the model; (2) Evaluate the model by comparing the changes of the path, velocity and acceleration predicted by the model with the actual observation. The higher the coincidence degree of the image, the better the model performance.

In this paper, particle swarm optimization algorithm [28,29] is selected as the optimization algorithm of I-IDM model parameter calibration, genetic algorithm is selected as the optimization algorithm of I-Wiedemann99 model parameter calibration, and speed and headway are selected as performance indexes.

## 3.1 Following event extraction

In recent years, the modeling of follow – through model has attracted more and more attention from scholars. In this section, based on the current classical following model theory, an appropriate following model is selected for parameter calibration under different rainfall intensity in combination with the research content of this paper, and 40 effective following teams are sorted out by Sumo simulation data as the basic data set for parameter calibration of the following model.

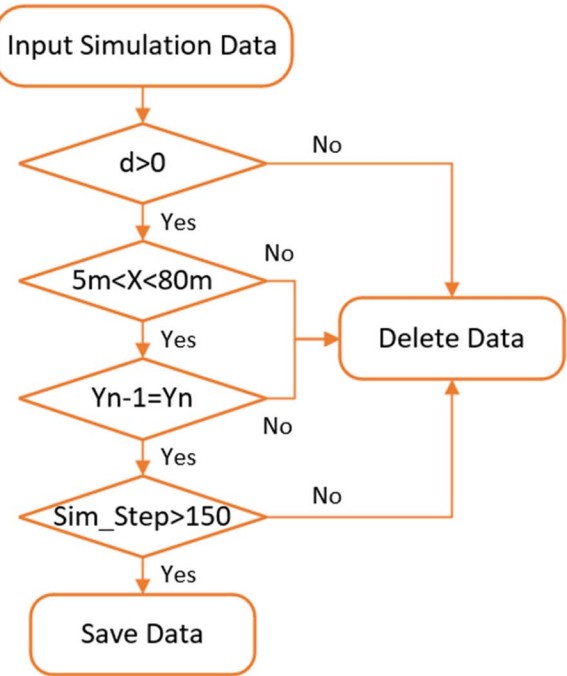

According to Zhu et al. 's research, it is concluded that the following events should meet the criteria such as keeping the following vehicle in the same lane as the front target and keeping the following vehicle in a stable state [30]. The extraction rules for effective following fragments are as follows: (1) Ensure that there is a vehicle in front of the following vehicle and maintain a certain distance from the lead vehicle $d > 0$; (2) remove the overcrowded or free flow traffic state, that is, the distance between the following vehicle and the lead vehicle; $5m < X < 80m$; (3) ensure that the following vehicle and the lead vehicle are in the same lane, that is; $Y_n = Y_{n-1}$; (4) ensure that the following vehicle and the lead vehicle have a long time before the stable following state, that is $Sim\_Step > 150 \, frame$ (more than 15 seconds). After screening, 40 valid following events are finally extracted. The extraction process of following events is shown in Fig 5.

This paper adopts the method of simulation verification to study. Considering that the state of traffic flow is different under different rain intensity, and since Sumo cannot directly simulate rain weather, this paper modified the parameters of Sumo rou file to replace the traffic flow under different rain intensity. Finally, Sumo Traci interface was used to output the vehicle speed, position and acceleration data of each frame in the simulation process. And automatically generate a.CSV file. The resulting csv file format is shown in Table 5:

### 3.2 Follow-up event analysis

The effective car-following events extracted in this paper include 8 data including the changes of position, distance, velocity and acceleration between Target Veh and Preceding Veh. One of the car-following pairs is shown in Fig 6:

As can be seen from Fig 6, the greater the following distance, the greater the acceleration of the following vehicle, and the greater the expectation of speed; On the contrary, when the distance between the following vehicle and the lead vehicle is too close, the following vehicle will choose to slow down to maintain a safer workshop distance, and the trend of the following vehicle speed curve is basically the same as the trend of the workshop distance. It can be seen from the above that the following events extracted in this paper are in line with objective laws, and are suitable for parameter calibration

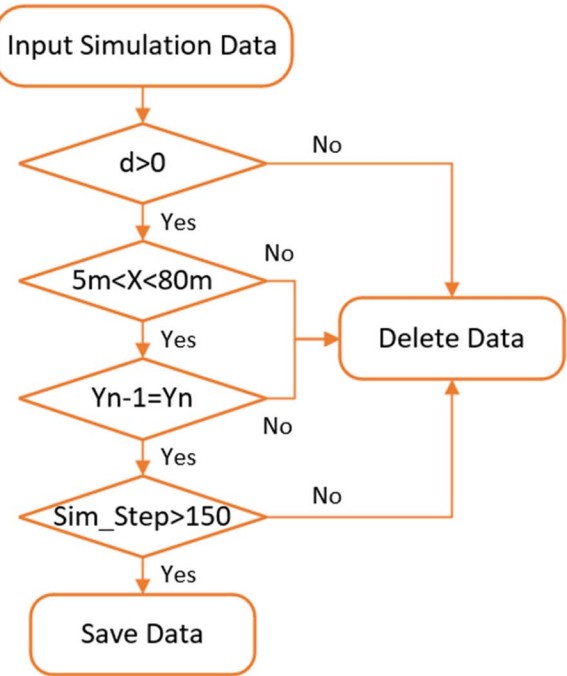

**Fig 5. Process of extracting follow events.**

**Table 5. Data set formats.**

| Parameter names | Meaning | Units |
|---|---|---|
| **Main_Pos** | Following vehicle position | m |
| **Main_Spe** | Following vehicle speed | m/s |
| **Main_Acc** | Following vehicle acceleration | $m/s^2$ |
| **Rela_Pos** | Relative distance | m |
| **Lead_Pos** | Lead vehicle position | m |
| **Lead_Spe** | Lead vehicle speed | m/s |
| **Lead_Acc** | Lead vehicle acceleration | $m/s^2$ |
| **Sim_Step** | Emulating step size | 0.1 s |
| **Main_ID** | Following vehicle ID | # |
| **Leader_ID** | Lead vehicle ID | # |

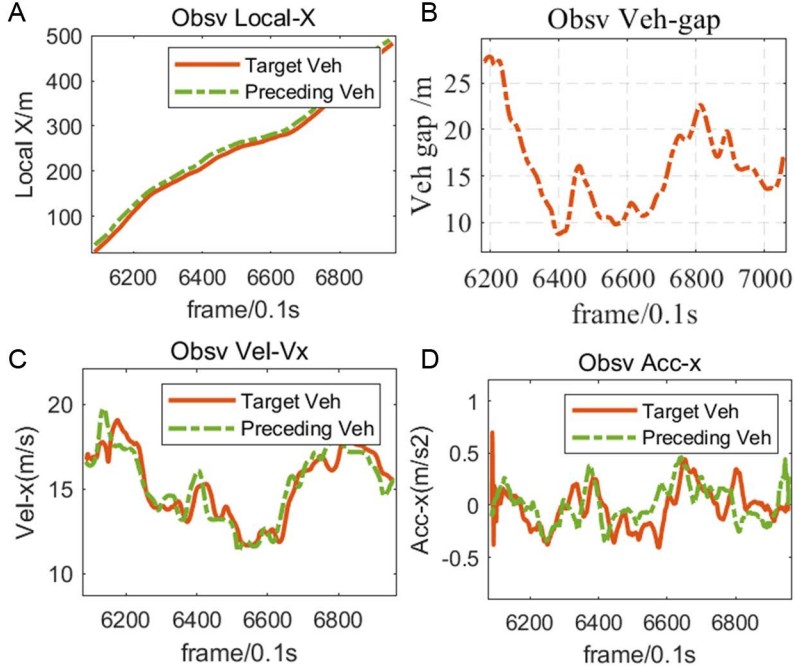

**Fig 6. Example vehicle following event.**

of the following model. The details of the difference between speed and acceleration of following pair under different rain intensity are shown in Table 6:

It can be seen from Table 6 that the standard deviation of vehicle speed and acceleration show a decreasing trend in the process of increasing rain intensity, the difference between the maximum and minimum speed decreases from 19.36m/s to 15.29m/s, and the acceleration range of the vehicle decreases continuously, which further indicates that the value of acceleration and deceleration of the vehicle is smaller and more concentrated with the increase of rain intensity. The expected value of the driver's change speed becomes lower, and this trend increases with the increase of the rain intensity, indicating that the above data are consistent with the psychology of drivers in rainy days and the objective fact that drivers are more cautious when driving in rainy days and the heavier the rain intensity, the more cautious the drivers are.

**Table 6. Details of following pair data under different rain intensities.**

| Weather Conditions | Items | Standard deviation | Minimum | Maximum |
|---|---|---|---|---|
| Clear | Speed(m/s) | 6.21 | 18.89 | 38.25 |
| | Acceleration (m/s$^{-2}$) | 2.29 | −2.74 | 3.16 |
| Light rain | Speed (m/s) | 5.15 | 18.05 | 33.68 |
| | Acceleration(m/s$^{-2}$) | 2.06 | −2.53 | 3.01 |
| Moderate rain | Speed (m/s) | 3.89 | 15.68 | 30.26 |
| | Acceleration (m/s$^{-2}$) | 1.79 | −2.22 | 2.37 |
| Heavy Rain | Speed (m/s) | 3.03 | 10.25 | 25.54 |
| | Acceleration (m/s$^{-2}$) | 1.62 | −2.05 | 2.21 |

### 3.3 Parameter calibration of I-IDM model considering micrometeorological information

**3.3.1 Calibration process of I-IDM model based on particle swarm optimization algorithm.** In the calibration of I-IDM follow-through model, particle swarm optimization algorithm has many advantages. Particle swarm optimization algorithm can quickly search the optimal parameter combination of I-IDM model, so that the I-IDM model can describe the following behavior more accurately. The accuracy and reliability of traffic simulation model can be improved by using particle swarm optimization algorithm to calibrate parameters of I-IDM model.

In this paper, 40 following fragments were selected and screened from Sumo simulation data set, and calibration was carried out several times. The basic process of particle swarm algorithm is shown in Fig 7. The parameters of PSO particle swarm optimization were set as shown in Table 7:

**3.3.2 Calibration results of I-IDM model based on particle swarm optimization algorithm.** After optimization and solution by particle swarm optimization algorithm, calibration results of parameters of the I-IDM model under normal weather and different rainfall intensities are shown in Table 8. The average values of calibration of 40 groups of following pairs are selected as calibration results.

In the I-IDM model, the maximum acceleration $\alpha_{max}^{(n)}$ reflects the sensitivity of the following driver to his own desired speed and desired spacing when making acceleration decisions in the car-following behavior. The expected acceleration and deceleration represent the driver's own comfortable acceleration/deceleration in the following process. Specifically, the comfortable deceleration refers to the vehicle decelerating or accelerating in a comfortable way when the vehicle meets the speed change of the vehicle in front, so that the passengers in the vehicle feel comfortable and will not produce a sense of severe jolting. It can be seen from Table 4-4. Regardless of whether it is raining or not, drivers have a tendency to accelerate. However, different from sunny days, drivers prefer to complete the acceleration process with a small acceleration and reach their desired speed in rainy days. The expected speed $\tilde{v}_n$ represents the maximum speed expected by the driver behind. With the continuous increase of rain intensity, the expected speed of the driver behind decreases continuously, which indicates that the driver is more cautious in the consideration of his own life safety when driving in rainy days. The expected headway $\widetilde{T}_n(t)$ represents the expected time distance between the following vehicle and the lead vehicle in the following process. With the increase of rain intensity, $\widetilde{T}_n(t)$ increases significantly, which indicates that the driver expects to keep a longer distance from the lead vehicle in rainy conditions. With the increase of rain intensity, the blocking distance $S_{jam}^{(n)}$ also increases at rest. In addition, as $\lambda$ increases with the increase of rain intensity, the maximum acceleration, expected acceleration and expected speed all decrease, and the expected headway is also increasing, which conforms to the objective law and proves the rationality of the calibration results. Examples of calibration results for 40 car-following pairs are shown in Fig 7. In the figure, the speed change of the following vehicle after calibration is basically consistent with that before calibration, and the distance between vehicles also has a good calibration effect.

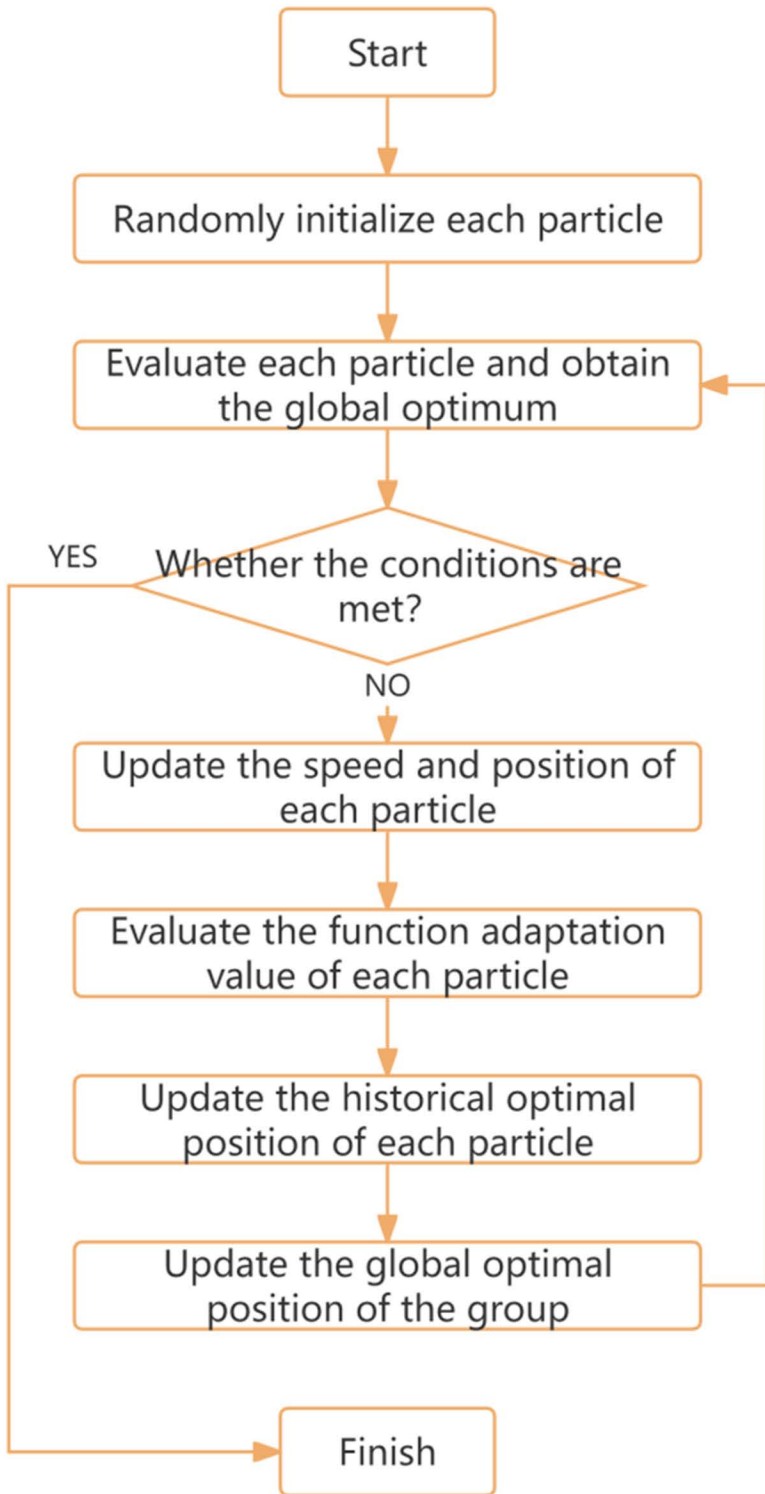

**Fig 7. Flowchart of the particle swarm optimization algorithm.**

**Table 7. Particle swarm optimization (PSO) parameter Settings.**

| Parameter Name | Function name | Parameter Settings |
|---|---|---|
| **Minimum particle speed** | v_min | 0.4 |
| **Maximum particle speed** | v_max | 0.9 |
| **c1 Learning factor** | c1_syms | [2.5,0.50] |
| **c2 learning factor** | c2_syms | [1.0,2.55] |
| **dimensionality** | PosDim | 5 |
| **Population size** | pop_size | 300 |
| **Number of iterations** | iter_num | 800 |
| **Allowed convergence error** | eps_val | 10^(−6) |

**Table 8. Calibration results of 4I-IDM model parameters.**

| Parameters | Meaning | Units | Varying rain intensity | | | |
|---|---|---|---|---|---|---|
| | | | No rain | Light Rain | Moderate rain | Heavy rain |
| $\alpha_{max}^{(n)}$ | Maximum acceleration | m/s² | 3.81 | 3.46 | 2.65 | 1.85 |
| $\beta$ | Desired acceleration | m/s² | 4.21 | 3.91 | 3.91 | 2.31 |
| $\widetilde{v}_n$ | Desired speed | m/s | 31.32 | 28.45 | 25.48 | 21.86 |
| $\widetilde{T}_n(t)$ | Expected headway | s | 1.86 | 2.07 | 2.51 | 3.46 |
| $S_{jam}^{(n)}$ | Block spacing at rest | m | 2.89 | 4.57 | 5.56 | 6.05 |
| $\lambda$ | Factors influencing the driver's judgment of road conditions | # | 0.6 | 0.7 | 0.8 | 0.9 |

As shown in Fig 8, Observe-Path and Cali-Path-PSO are the traveling paths of the following vehicles before and after calibration, and Observe-gap and Cali-gap-PSO are the relative positions of the following vehicles and the lead vehicles before and after calibration. Observe-Vel and Cali-Vel-PSO are the changes in the speed of the vehicles behind and after the calibration. The driving data generated by the calibrated following model, such as the distance, relative position and speed between the following vehicle and the lead vehicle, especially the driving path of the following vehicle before and after calibration, are highly consistent with the observed data, which proves the rationality of using particle swarm optimization algorithm to calibrate the I-IDM model. It can be clearly seen from the figure that the simulation data of the I-IDM model after calibration is basically consistent with the observed data of the following vehicle before calibration.

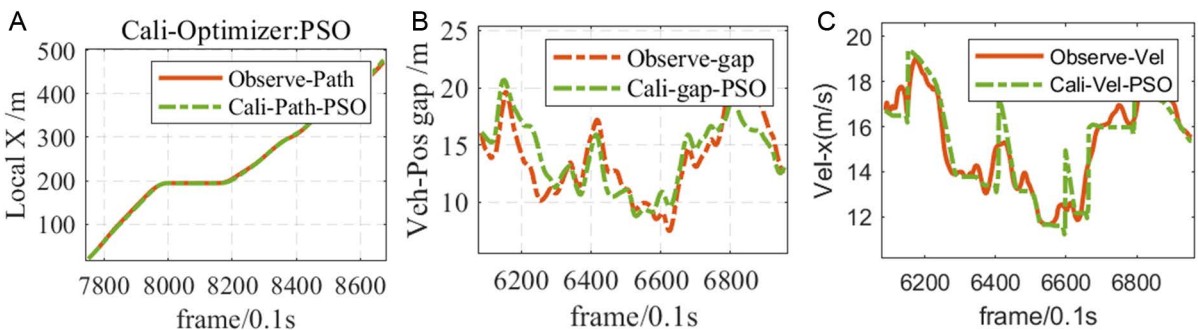

**Fig 8. Comparison of I-IDM parameters after calibration and before and after calibration.**

After 100 iterations of parameter calibration of I-IDM model using particle swarm optimization algorithm, the E-RMSPE change curve tends to be smooth and does not fall into local optimum, and the result converges. The change curve of E-RMSPE for IDM model parameter calibration is shown in Fig 9.

To sum up, the calibration of I-IDM model parameters by particle swarm optimization in this paper is consistent with objective laws, and can better describe the changes of drivers' driving behavior under different rain intensities.

### 3.4 Parameter calibration of I-Wiedemann99 model considering micrometeorological information

**3.4.1 Calibration process of I-Wiedemann99 model based on genetic algorithm.** The calibration of the I-Wiedemann99 model also takes speed and headway as performance indicators, and RMSPE as the objective function. The execution process of genetic algorithm is shown in Fig 10. The parameters to be calibrated and their value ranges are shown in Table 9:

**3.4.2 Calibration results of I-Wiedemann99 model based on genetic algorithm.** Since the I-Wiedemann99 model requires a large number of parameters to be calibration, and the optimization effect of particle swarm optimization algorithm is poor when it is optimized by experiments. In addition, because genetic algorithm is not easy to be trapped in the local optimal solution, this paper selects genetic algorithm (GA) as the optimization algorithm of I-Wiedemann99 model.

In this paper, we use MATLAB genetic algorithm toolbox to achieve iterative optimization of genetic algorithm. Due to the randomness of genetic algorithm, in order to avoid falling into the local optimal solution, this paper calibrates each of 40 vehicle following pairs 10 times, and takes the result with the smallest error as the final result. In addition, since the I-Wiedemann99 model requires a large number of calibration parameters, and in order to reduce the error of calibration results, the population size and iteration times of parameter calibration for the I-Wiedemann99 model set in this paper are larger than the number of calibrations for the I-IDM model. Table 10 lists the specific parameters to be set.

After optimizing the default following model by genetic algorithm, the calibration results of I-Wiedemann99 model considering different rainfall intensity of micro weather strips are shown in Table 11, and the average values of 40 following pairs are also selected as the final calibration results.

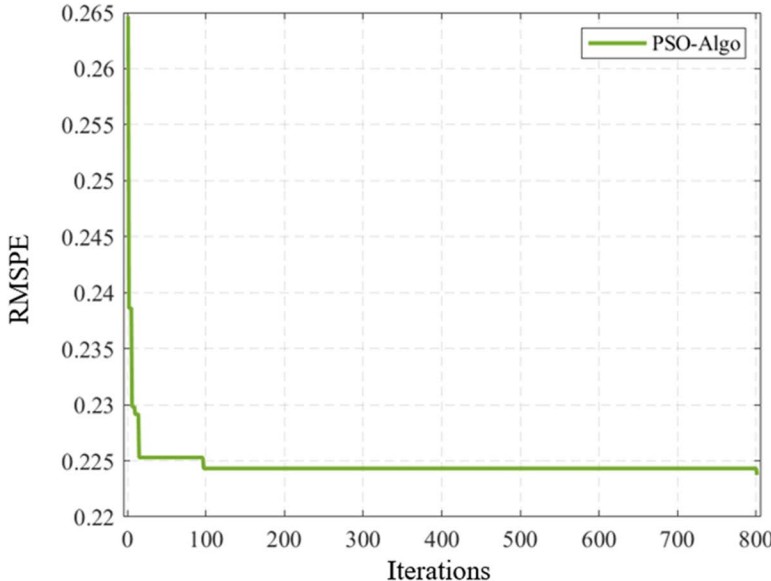

**Fig 9. Change curve of E-RMSPE for IDM model parameter calibration.**

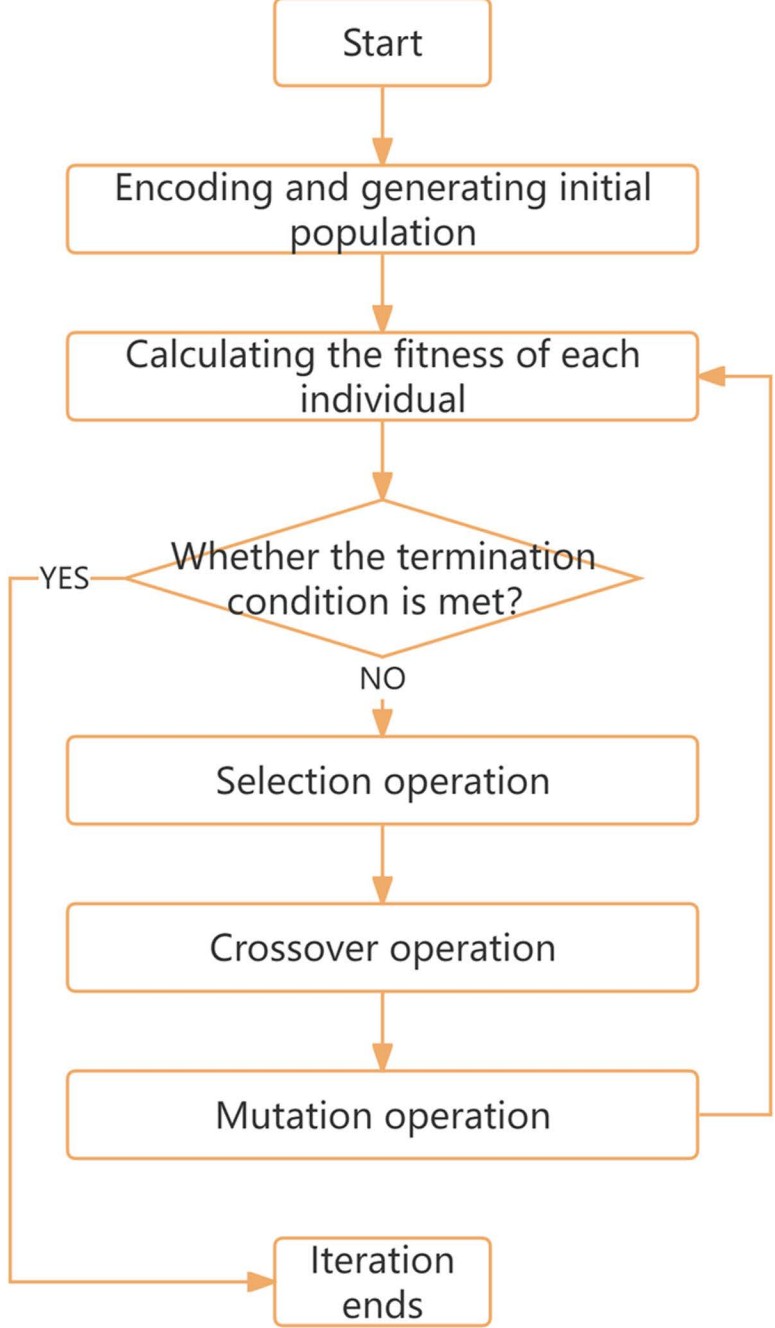

**Fig 10. Flowchart of the particle swarm optimization algorithm.**

It can be seen from the calibration results that in the I-Wiedemann99 model, with the increase of rain intensity, the expected stopping distance and the expected time headway are increasing. The value of CC2 decreases continuously; The driver of the following vehicle is more sensitive to changes in the speed of the vehicle in front; The expectation that the driver behind has a certain acceleration regardless of the weather conditions when the desired speed is not reached; The expected speed of the driver of the following vehicle is constantly decreasing, and the driver has the expectation of

**Table 9. Calibration parameters and their value ranges for the I-Wiedemann99 model.**

| Parameters | Meaning | Units | Range of values |
|---|---|---|---|
| CC0 | Parking expectation spacing | m | [100,200000] |
| CC1 | Time headway | s | [0.1, 5] |
| CC2 | Following variables | m | [0.1, 10] |
| CC3 | Enter the "following" status threshold | s | [- 20, 0.1] |
| CC4 | The "following" state threshold where the speed of the following vehicle is less than the speed of the lead vehicle | m/s | [-5, -0.1] |
| CC5 | The "following" state threshold at which the speed of the following vehicle is greater than the speed of the lead vehicle | m/s | [0.1, 5] |
| CC6 | Concussion speed | rad/s | [0.00001, 0.002] |
| CC7 | Concussive acceleration | m/s | [- 1, 1] |
| CC8 | Stopping acceleration | m/s | [0.1,8] |
| CC9 | Desired acceleration at 22.22m/s | m/s | [0.1,8] |
| $V_0$ | Desired speed | m/s | [1,100] |

**Table 10. Parameters of the genetic algorithm.**

| Genetic Algorithm Parameters | Settings | Description |
|---|---|---|
| Population type | Floating-point vector | each individual represents a real-valued vector |
| Population size | 500 | Number of individuals in each generation of the population |
| Method of generation of offspring | Fitness sorting, simulation variation | The fittest individuals from the parent generation are directly inherited by the offspring; the remaining offspring are generated through simulation mutation. |
| Mutating operations | Constraint correlation | Random variation in the range of parameters set |
| Simulation operation | Binary Crossover | 0-1 Crossover |
| Maximum number of iterations | 1500 | Upper bound of iterative algebra |
| Allowable convergence error | 1 x 10 ^ (−6) | Determine if the optimal solution converges |

**Table 11. Calibration results of I-Wiedemann99 model parameters.**

| parameters | Different rain intensities | | | |
|---|---|---|---|---|
| | No rain | Light rain | Moderate rain | Heavy rain |
| CC0 | 8.04 | 8.3 | 9.56 | 11.07 |
| CC1 | 1.03 | 1.53 | 1.72 | 1.96 |
| CC2 | 0.64 | 0.64 | 0.04 | 0.02 |
| CC3 | −9.79 | −8.58 | −3.7 | −3.97 |
| CC4 | −1.5 | −1.06 | −0.23 | −0.09 |
| CC5 | 1.5 | 1.06 | 0.23 | 0.09 |
| CC6 | 12.16 | 18.73 | 13.67 | 2.61 |
| CC7 | 0.24 | 0.34 | −0.16 | −0.82 |
| CC8 | 3.78 | 2.86 | 2.19 | 1.16 |
| CC9 | 2.49 | 1.91 | 1.46 | 0.45 |
| $V_0$ | 35.57 | 32.95 | 26.45 | 22.22 |
| λ | 0.6 | 0.7 | 0.8 | 0.9 |

driving at a higher speed under different conditions of rain intensity. The above results show that the improved model is consistent with the objective law and can describe the following behavior better.

As shown in Fig 11, Observe-Path and Cali-Path-GA are the traveling paths of the following vehicles before and after calibration, and Observe-gap and Cali-gap-GA are the relative positions of the following vehicles and the lead vehicles before and after calibration. Observe-Vel and Cali-Vel-GA are the changes in the speed of the vehicles behind and after the calibration. The driving data generated by the calibrated following model, such as the distance, relative position and speed between the following vehicle and the vehicle in front, are highly consistent with the observed data, which proves the rationality of using genetic algorithm to calibrate I-Wiedemann99. It should be noted that in 6400~6600 frames, the calibrated I-Wiedemann99 model can accurately reproduce the traffic behavior in which the distance between the two vehicles is too small and there is a large slowdown.

In summary, the calibration results of I-Wiedemann99 model parameters by using genetic algorithm accord with the objective following law, and the calibration effect is good. After 400 iterations of parameter calibration for I-Wiememann99 model using genetic algorithm, the change curve of E-RMSPE tends to be flat, and the results are convergent without falling into local optimal. The E-RMSPE Variation Curve for I-Wiedemann99 Model Parameter Calibration is shown in Fig 12.

### 3.5 Model verification and evaluation

**3.5.1 Analysis and evaluation of error of calibration model.** Calibrating the I-IDM and I-Wiedemann99 car-following models, the simulation verification method is used to verify the model to observe whether the calibrated model can accurately reflect the actual driving behavior and traffic flow conditions, so as to improve the prediction ability and application value of the model. In the process of establishing the car-following model in this paper, a total of 40 groups of car-following data collected through simulation experiments are selected. The average of the final results is shown in Fig 13:

It can be seen that for both I-IDM and I-Wiedemann99 models, the average error of calibration results increases with the increase of rain intensity. The reason is that drivers' psychology and road conditions in rainy days become more complex than those in no-rain conditions, which makes it difficult for the models to accurately reflect vehicle following states. In addition, when the two models were compared under different rainfall intensities, it was found that the average error of the I-Wiedemann99 model was larger than that of the I-IDM, especially when the rain intensity was larger, and the 85% quantile of the I- Wiedemann99 model was close to 0.3. Compared with the I-Wiedemann99 model, the error of the I-IDM model is smaller when the rain intensity is larger. The reason is that there are many parameters of I-Wiedemann99, and most of the parameters are not clear in practical significance, so it is difficult to accurately simulate the real vehicle following behavior, and eventually lead to large calibration error. The final results show that the I-IDM model has a good performance in both sunny weather and rainy weather. The model has a small error and can simulate the characteristics of traffic flow in rain environment.

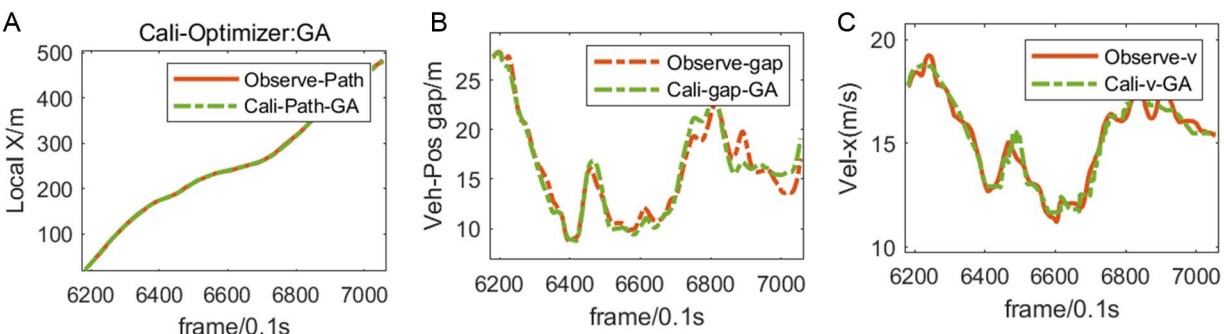

**Fig 11. Comparison of following vehicle conditions before and after I-Wiedemann99 parameter calibration.**

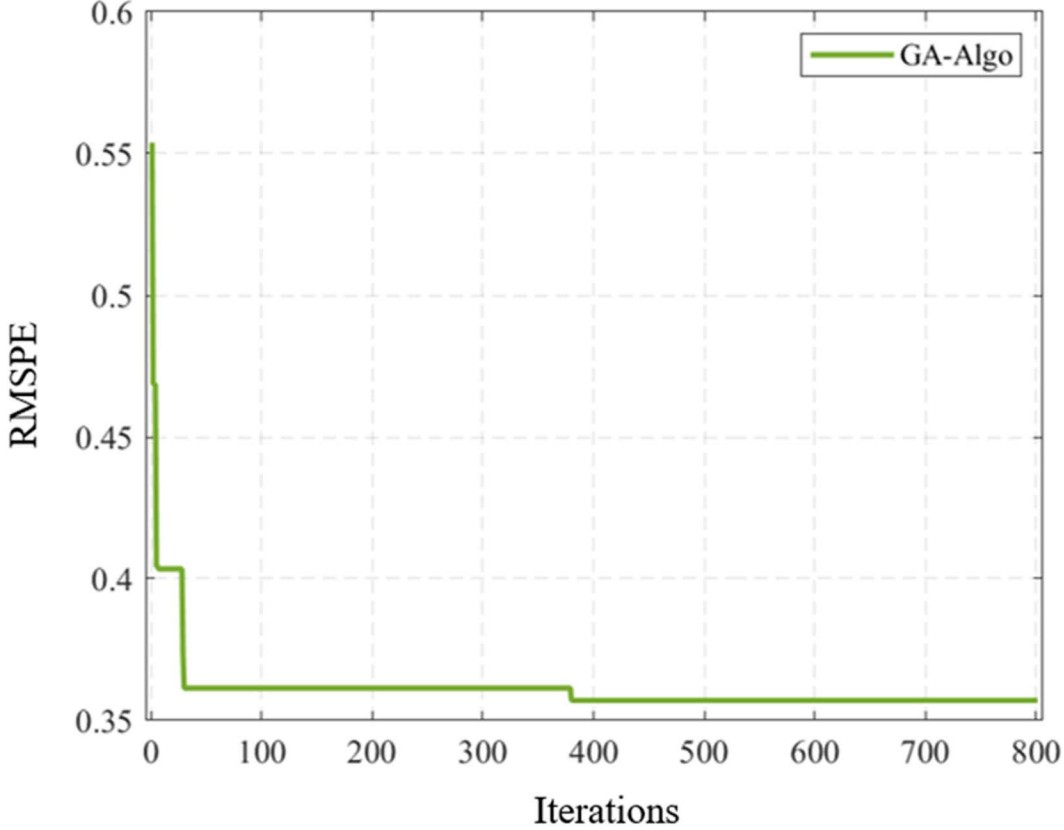

**Fig 12. E-RMSPE variation curve for I-Wiedemann99 model parameter calibration.**

In the case of no rainfall, the 85-quantile RMSPE cumulative frequency distribution after parameter calibration of the I-IDM model reached 0.44, that is, 85% RMSPE error was less than 0.44,and the 85-quantile cumulative frequency distribution of the RMSPE after parameter calibration of the I-Wiedemann99 model reached 0.38.That is, 85%of the RMSPE error in the calibration results is less than 0.38;In the case of light rain, the 85th quantile of RMSPE cumulative frequency distribution after calibration of I-IDM model parameters reached 0.39,and the 85th quantile of RMSPE cumulative frequency distribution after calibration of I-Wiedemann99 model parameters reached 0.35. In the case of moderate rain, the 85-quantile cumulative frequency distribution of RMSPE after calibration of I-IDM model parameters reaches 0.38, and the 85-quantile cumulative frequency distribution of RMSPE after calibration of I-Wiedemann99 model parameters reaches 0.30. In the case of heavy rain, the 85-quantile cumulative frequency distribution of RMSPE after calibration of I-IDM model parameters reaches 0.33, and the 85-quantile cumulative frequency distribution of RMSPE after calibration of I-Wiedemann99 model parameters reaches 0.26. In summary, the calibration and verification errors of the improved model parameters are within the acceptable range, and the calibrated model has a good predictive effect on the following behavior.Variation curve of the 85th percentile RMSPE for I-IDM and I-Wiedemann99 under different rain intensity levels is shown in Fig 14:

According to the change curve of the average verification error and the change curve of the 85-quantile, it can be seen that the calibration result of the I-IDM model becomes stable earlier than that of the I-Wiedemann99 model, indicating that the calibration result of the I-IDM model is better. After 40 calibration times of I-IDM model and I-Wiedemann99 model, the results areas follow: (1)the mean error and standard deviation of I-IDM model are smaller than that of I-Wiedemannn99

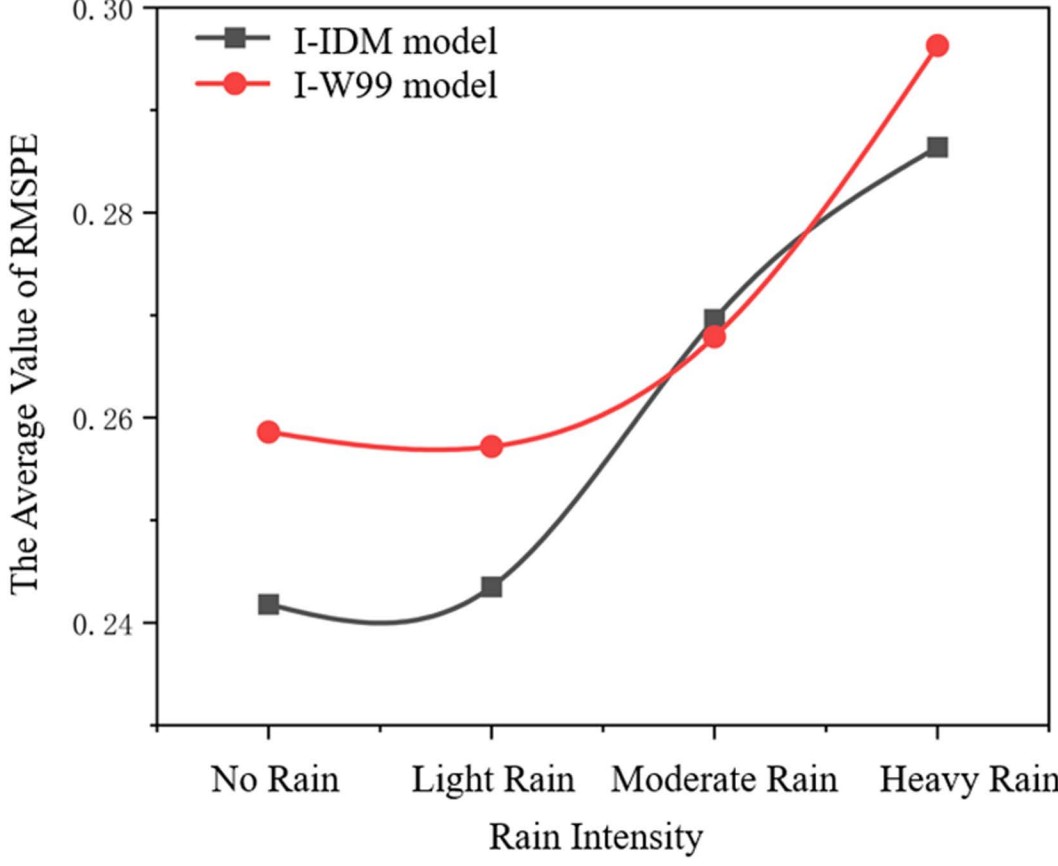

**Fig 13. Variation of the average validation error in following vehicle model calibration.**

model; (2)the maximum value of the RMSPE of the I-IDM model is 0.4613, and the minimum value is 0.1376. The maximum value of the RMSPE of the I-Wiedemann99 model is 0.4568, and the minimum value is 0.1324; (3)the parameter calibration results of I-Wiedemann99 model are more discrete than those of I-IDM model, and the I-IDM model calibrated in this paper is more consistent with the real vehicle following scene; (4)Errors of both the I-IDM model and I-Wiedemann99 model are increasing with the increasing of rain intensity. The 85-quantile of the I-Wiedemann99 model is only 0.26 under heavy rain, which is the largest calibration error of the I-IDM and I- Wiedemann99 models under the four kinds of rain intensity.

As shown in Fig 15 and Fig 16, it can be seen from the above figures that the cumulative distribution frequency of RMSPE in both I-IDM model and I-Wiedemann99 model is the earliest to reach 100% in the case of increasing rain intensity. Due to the increasing complexity of car-following behavior under different rain intensities, the cumulative distribution frequency of RMSPE of the two models reaches 100% later and later. In general, the parameter calibration errors of the improved I-IDM model and I-Wiedemann99 model are within an acceptable range, and the calibrated model is effective.

**3.5.2 Simulation realization and verification of the calibration model.** As to whether it is necessary to consider the factors affecting drivers' judgment of road conditions under the influence of micrometeorological conditions in the following model, the effectiveness of the model should be verified by comparing the average road speed, road loss time and vehicle speed stability in the simulation process of the model before and after the improvement. The specific simulation

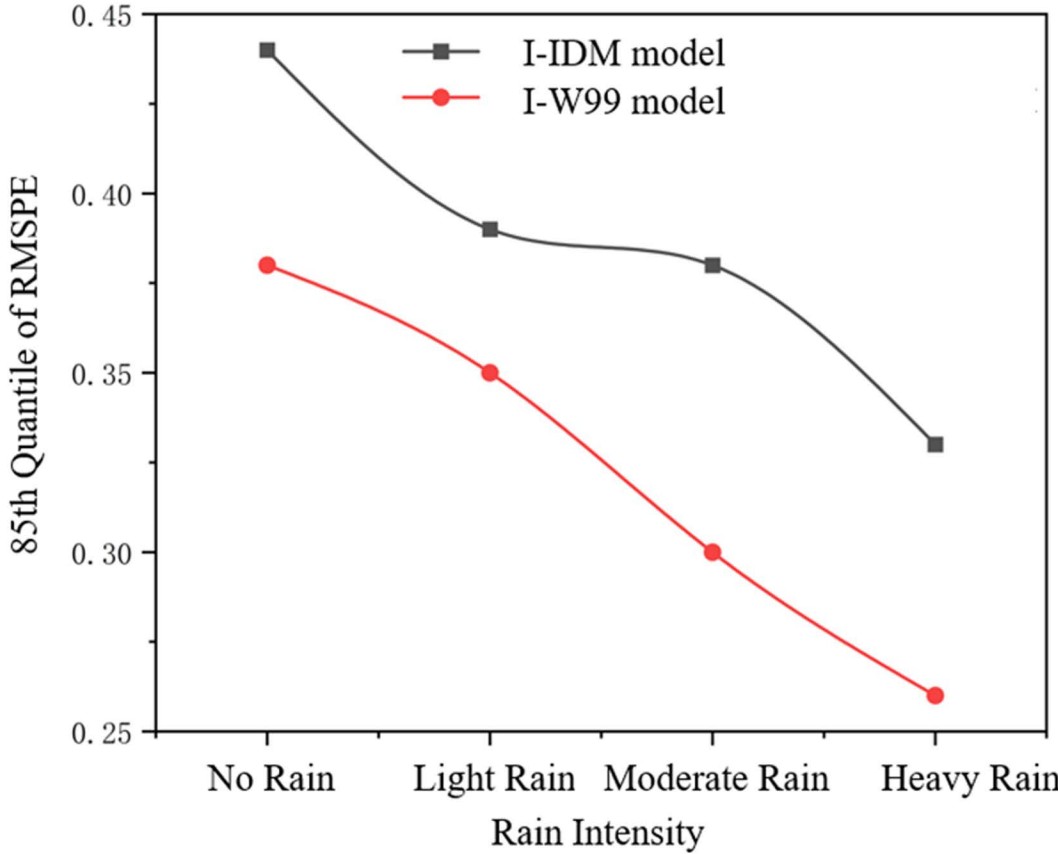

**Fig 14. Variation curve of the 85th percentile RMSPE for I-IDM and I-Wiedemann99 under different rain intensity levels.**

verification scheme is as follows. Since this paper mainly studies the characteristics of vehicle following behavior under different rain intensity, in order to avoid excessive lane change resulting in unstable traffic flow in this experiment, which will have an impact on the experiment, this paper establishes a closed-loop simulation scenario as shown in Fig 17. The basic simulation section selects a three-lane rectangular closed-loop road including ordinary straight sections, with a total length of4km. In this way, the influence of incoming vehicles on the following behavior can be ignored, which makes the research results more meaningful for reference.

The simulation verification process is set as follows: (1) Enter the default number of vehicles 200,the vehicle length is 5m,in which the default number of vehicles in the I-IDM model and the I-Wiedemann99 model are 100 vehicles each, the simulation running time is 1200s,and the simulation step length is 0.1s,to obtain the vehicle data in the middle section of the road in a relatively stable state;(2)Enter the number of vehicles 200,wherein the improved I-IDM model and I-Wiedemann99 model have 100 vehicles each, the vehicle length is 5m,the simulation running time is 1200s,the simulation step is 0.1s,and the vehicle data in the middle section of the relatively stable state is obtained. Record the average road speed and average loss time of drivers considering the influence of micro-meteorological conditions under the influence of micro-meteorological conditions, and then analyze the influence of drivers' judgment factors on road conditions under the influence of micro- meteorological conditions on traffic efficiency. The Influence of drivers' judgments of road conditions under micro meteorological conditions on average vehicle speed and average road loss time is shown in Fig 18 and Fig 19.

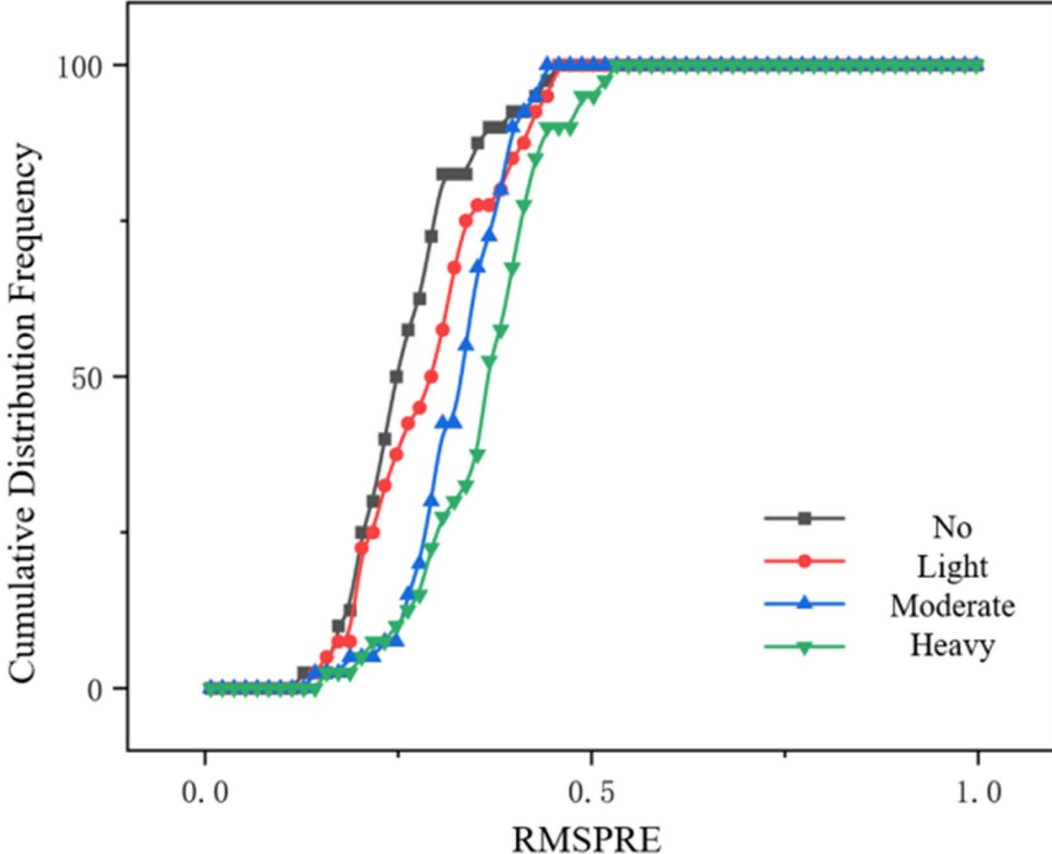

**Fig 15.  Cumulative frequency distribution of I-IDM model under different rainfall intensities.**

As can be seen from the figure, after considering the driver's judgment on road conditions under the influence of micrometeorological conditions, the overall average road speed is low, while the overall road loss time is high. This is because this paper believes that the driver's judgment on road conditions under the influence of micrometeorological conditions directly affects the driver's estimation of the motion state of the vehicle in front of him during driving. Under different rain intensity conditions, drivers' judgment of road conditions will be affected to different degrees, and they always think that they need to maintain a lower speed or a larger distance to ensure safety, which leads to a decrease in the average road speed and an increase in the lost time due to the failure to meet the expected speed. It can be seen from the figure above that the model is basically consistent with the actual situation after adding drivers' judgment factors on road conditions under the influence of micro-meteorological conditions, which reflects the effectiveness of the models.As shown in Fig 20. To integrate the above achievements into existing practices, it is necessary to sys-tematically upgrade data collection, develop automated toolchains, and achieve in-depth collaboration with simulation platforms.

## 4.  Conclusion

This paper studies the parameter calibration methods of car-following models and proposes a calibration framework for the I-IDM model based on Particle Swarm Optimization and the I-Wiedemann99 model based on Genetic Algo-rithms. The framework compares simulation results from calibrated data and measured data, demonstrating that

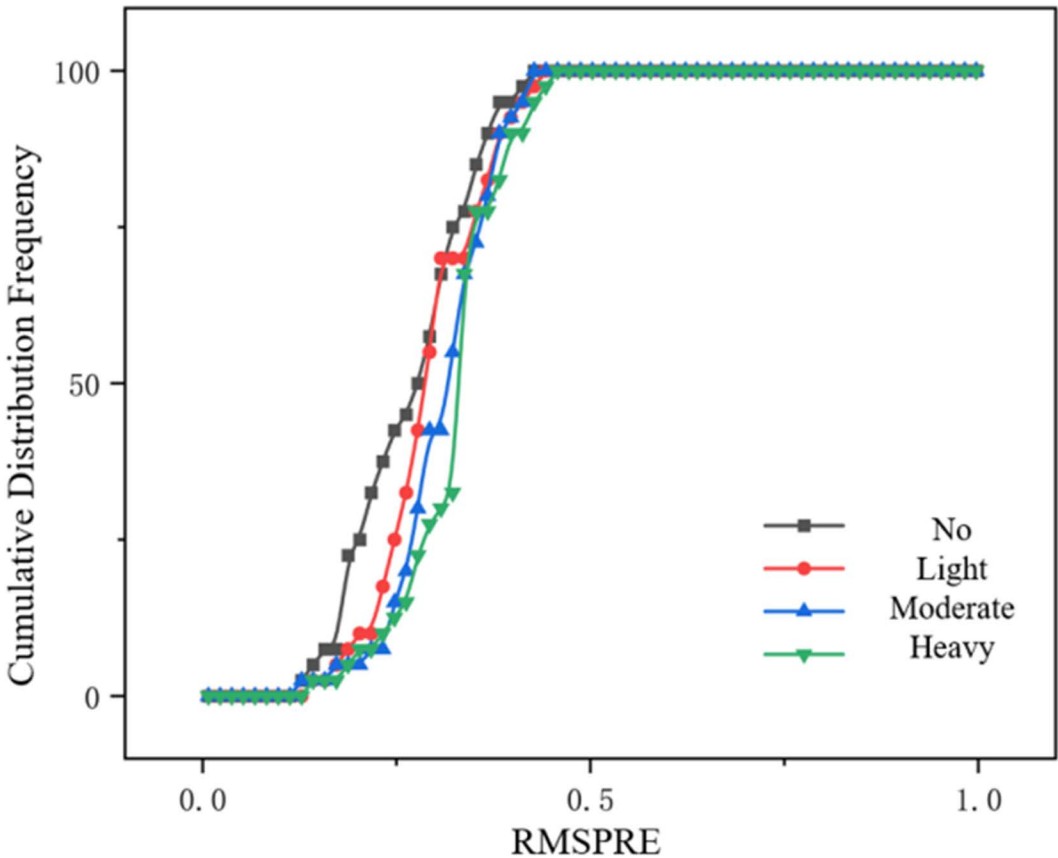

**Fig 16. Cumulative frequency distribution of I-Wiedemann99 model under different rainfall intensities.**

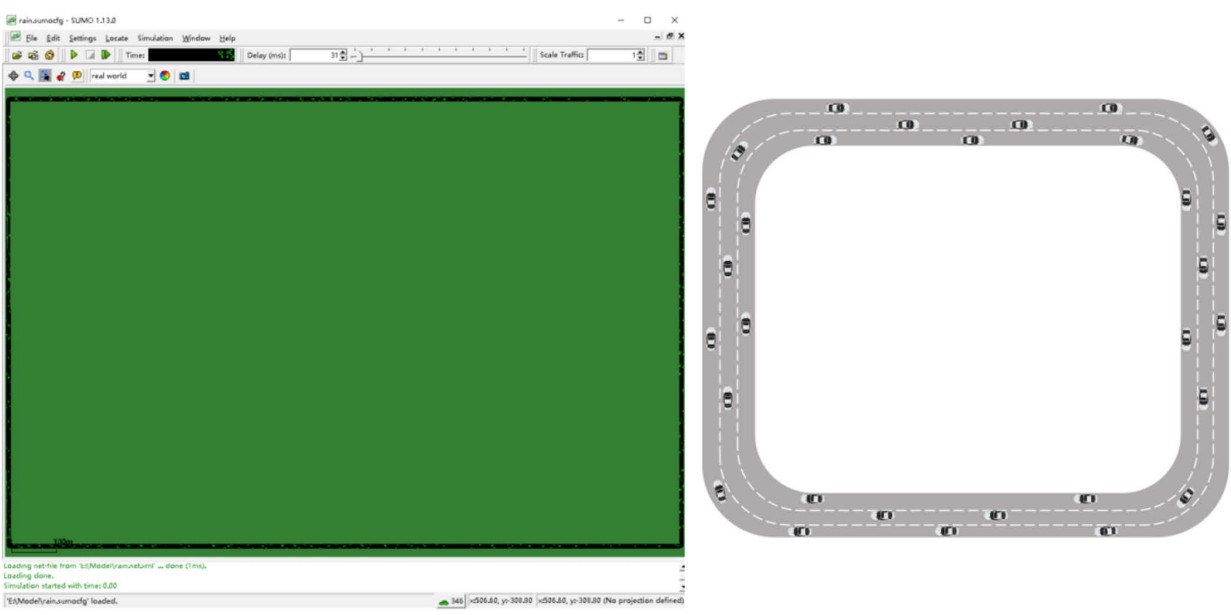

**Fig 17. Schematic diagram of the closed-loop simulation scenario.**

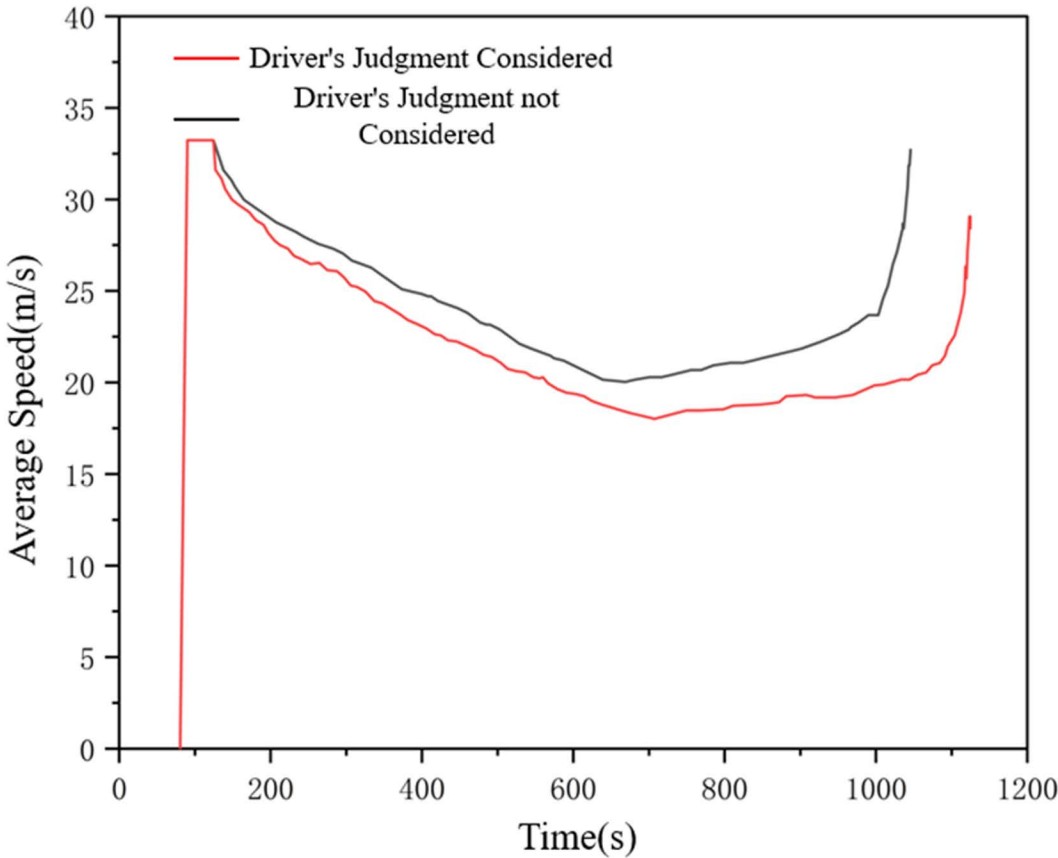

**Fig 18. Influence of drivers' judgment of road conditions under micrometeorological conditions on average road speed.**

both Particle Swarm Optimization and Genetic Algorithms can effectively search for global optimal values and can be applied to the calibration of simulated car-following data. Additionally, this paper introduces factors influencing drivers' judgment of road conditions under different micro-meteorological conditions into the default IDM and Wiedemann99 models, resulting in improved I-IDM and I-Wiedemann99 models. Parameter calibration is conducted, and simulations are carried out using both default parameters and calibrated parameters. The results show that the traffic flow simulated after parameter calibration is more stable, with slower variations in vehicle speed. Moreover, the cumulative distribution curves after two rounds of calibration indicate that the I-IDM model approaches stability earlier than the I-Wiedemann99 model. The calibrated I-IDM model better describes car-following behavior compared to the default IDM model, exhibiting lower error and aligning with the traffic flow characteristics in rainy environments. Finally, validation of the calibrated model through SUMO simulations proves its effectiveness and alignment with actual conditions. Although the calibrated car-following model can nearly perfectly replicate synthetic driving behavior, there are still discrepancies in the calibration experiments using real data. This may be because car-following models are based on general trends of vehicle operation rather than the specific behavioral details of individual drivers. Therefore, future research should conduct an in-depth analysis of the causes of these discrepancies to better calibrate the parameters of car-following models, enabling them to more accurately simulate driver behavior.

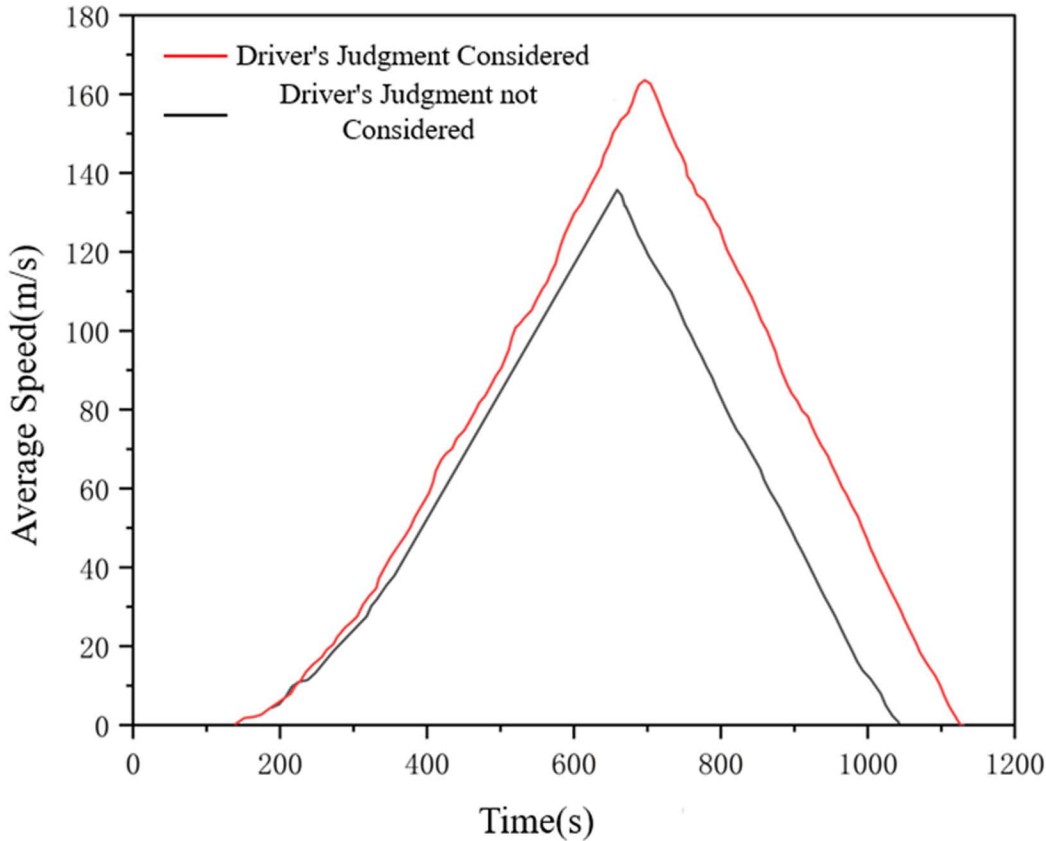

**Fig 19. Influence of drivers' judgment of road conditions under the influence of micrometeorological conditions on average road loss time.**

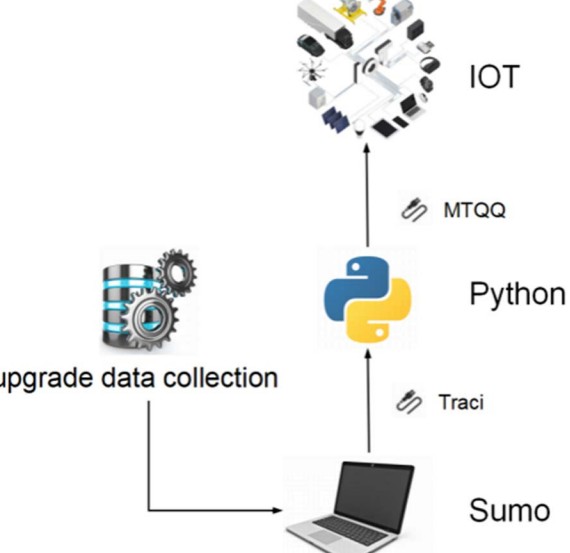

**Fig 20. The calibration methodology integrated into practices.**

# Supporting information

**S1 Data. Minimal dataset.**
(ZIP)

**S1 File. Original files.**
(ZIP)

# Author contributions

**Conceptualization:** Jian Ma, Liyan Zhang.

**Methodology:** Jian Ma, Zheng Qian.

**Software:** Zongwei Gao.

**Validation:** Liyan Zhang, Zheng Qian.

**Writing – original draft:** Jian Ma, Yuchen Zhang, Liyan Zhang, Zheng Qian.

**Writing – review & editing:** Yuchen Zhang, Zongwei Gao, Keyi Cao, Qianlong Fu.

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
