## [Decision Letter · Decision Letter 0]

Dear Dr. Zhang,

Thank you for submitting your manuscript to PLOS ONE. After careful consideration, we feel that it has merit but does not fully meet PLOS ONE’s publication criteria as it currently stands. Therefore, we invite you to submit a revised version of the manuscript that addresses the points raised during the review process.

We look forward to receiving your revised manuscript.

Kind regards,

Zhihong (Arry) Yao, Ph.D.

Academic Editor

PLOS ONE

The name of the colleague or the details of the professional service that edited your manuscript.

“This research was funded by Natural Science Foundation of Jiangsu Provincial(Y2020LX017). Construction System Project of Jiangsu Provincial,(2020ZD14�; Philosophy and Social Science Projects of Universities in Jiangsu Province(2023STYB1420); Postgraduate Research and Practice Innovation Program of Jiangsu Province(SJCX20_1117, SJCX21_1420, and KYCX21_2999); Social Science Foundation Project of Suzhou(Y2020LX025).”

5. We note that your Data Availability Statement is currently as follows: [All relevant data are within the manuscript and its Supporting Information files.]

Reviewers' comments:

Reviewer's Responses to Questions

**Comments to the Author**

1. Is the manuscript technically sound, and do the data support the conclusions?

Reviewer #1: Yes

Reviewer #2: Yes

2. Has the statistical analysis been performed appropriately and rigorously?

Reviewer #1: Yes

Reviewer #2: Yes

3. Have the authors made all data underlying the findings in their manuscript fully available?

Reviewer #1: No

Reviewer #2: Yes

4. Is the manuscript presented in an intelligible fashion and written in standard English?

Reviewer #1: Yes

Reviewer #2: Yes

Reviewer #1: The paper tried to addresses the problem of how micro-meteorological information can affect the calibration of traffic simulation models to better reflect driver following behavior. The topic is very interesting and I think the manuscript can benefit from the specific comments:

1. The references are really old, please update relevant papers.

2. The formatting of the equations is chaotic.

3. Data used in this paper should be described more clearly�such as the amount of data and the simulation section.

4. It is unclear that whether the authors conducted a field experiment. In conclusion, the authors stated that :"there are still discrepancies in the calibration experiments using real data". However, I did not see any discussion about the real data experiments.

Reviewer #2: The manuscript examines the Intelligent Driver Model (IDM) and the Wiedemann99 model, considering the influence of micrometeorological conditions. By incorporating a driver’s judgment factor, λ, the IDM and Wiedemann99 models are improved, leading to the development of new models: I-IDM and I-Wiedemann99. Overall, this manuscript is well-structured and composed in acceptable English. The paper addresses a significant topic that is likely to be of interest to the readership and may be considered for publication following revision. The author should meticulously improve the content of the paper in accordance with the review suggestions.

#1. It is recommended to reduce the conclusions in the abstract by incorporating only the most significant result from the proposed model calibration.

#2. Introduction part. There is a lack of references to existing related studies. It is challenging to ascertain the novelty of this paper without a comprehensive gap analysis supported by existing related research as background and to define the problem statement.

#3. The authors are kindly requested to provide a more detailed explanation of the methodology, specifically focusing on the calibration procedures that transform input data into calibration outcomes and subsequent validation. To facilitate better comprehension for the readership, it is suggested that a flowchart be incorporated, illustrating the complete sequence of calibration and validation steps.

#4. Figure 15. Quality of Figure 15 need to improve.

#5. A comprehensive analysis of the results should be presented, offering a more nuanced interpretation, and demonstrating how the calibration methodology can be integrated into existing practices.

#6. Evaluate the policy implications derived from the empirical evidence for China and other nations with analogous transportation systems.

**Do you want your identity to be public for this peer review?** For information about this choice, including consent withdrawal, please see our Privacy Policy

Reviewer #1: **Yes: ** Gen Li

Reviewer #2: No

---

## [Author Response · Author response to Decision Letter 1]

15 May 2025

We sincerely appreciate the reviewers for pointing out all the shortcomings, and we have made revisions in accordance with the journal's requirements.

Reviewer #1: The paper tried to addresses the problem of how micro-meteorological information can affect the calibration of traffic simulation models to better reflect driver following behavior. The topic is very interesting and I think the manuscript can benefit from the specific comments:

1.The references are really old, please update relevant papers.

Response: Some references have been updated. The references cited in the background section of this paper follow a certain chronological logic. Some of the classical theoretical models were indeed proposed quite some time ago. However, the authors believe that since we have drawn upon the wisdom of previous scholars, it remains necessary to retain these references.

2.The formatting of the equations is chaotic.

Response: The formatting of the equations have been updated. Now we use the formula editor built into Word to edit the formula.

3.Data used in this paper should be described more clearly�such as the amount of data and the simulation section.

Response: The data in the text, including table data, have been improved again.

4.It is unclear that whether the authors conducted a field experiment. In conclusion, the authors stated that :"there are still discrepancies in the calibration experiments using real data". However, I did not see any discussion about the real data experiments.

Response: The experimental data referenced here was provided by local traffic management department, hence it is not convenient to disclose. This paper presents preliminary attempts, and authors plan to further explore this issue in subsequent research.

Reviewer #2: The manuscript examines the Intelligent Driver Model (IDM) and the Wiedemann99 model, considering the influence of micrometeorological conditions. By incorporating a driver’s judgment factor, λ, the IDM and Wiedemann99 models are improved, leading to the development of new models: I-IDM and I-Wiedemann99. Overall, this manuscript is well-structured and composed in acceptable English. The paper addresses a significant topic that is likely to be of interest to the readership and may be considered for publication following revision. The author should meticulously improve the content of the paper in accordance with the review suggestions.

#1. It is recommended to reduce the conclusions in the abstract by incorporating only the most significant result from the proposed model calibration.

Response: The refinement of the abstract has been completed.

#2. Introduction part. There is a lack of references to existing related studies. It is challenging to ascertain the novelty of this paper without a comprehensive gap analysis supported by existing related research as background and to define the problem statement.

Response: We have added relevant discussions to establish the novelty of this paper. The introduction is divided into two sections—background and literature review—following a chronological logic. It provides a comprehensive analysis of the considerations in car-following model research from 1950 to 2020.

#3. The authors are kindly requested to provide a more detailed explanation of the methodology, specifically focusing on the calibration procedures that transform input data into calibration outcomes and subsequent validation. To facilitate better comprehension for the readership, it is suggested that a flowchart be incorporated, illustrating the complete sequence of calibration and validation steps.

Response: We have added Fig 7. Flowchart of the particle swarm optimization algorithm and Fig 10. Flowchart of the genetic algorithm.

#4. Figure 15. Quality of Figure 15 need to improve.

Response: After modification, it becomes Fig 16. On the basis of the simulation system operation demonstration, we have incorporated a schematic diagram of the closed-loop simulation scenario.

#5. A comprehensive analysis of the results should be presented, offering a more nuanced interpretation, and demonstrating how the calibration methodology can be integrated into existing practices.

Response: As shown in the newly added Fig20 , integrating the above achievements into existing practices requires systematic upgrades to data collection, development of automated toolchains, and in-depth collaboration with simulation platforms.

#6. Evaluate the policy implications derived from the empirical evidence for China and other nations with analogous transportation systems.

Response: These discussions will be elaborated by the authors in their next phase of research, and have currently been incorporated into the review section of the subsequent article to ensure continuity across our series of studies.

---

## [Decision Letter · Decision Letter 1]

Calibration of parameters in microscopic traffic flow simulation models considering micro-meteorological information

PONE-D-24-43241R1

Dear Dr. Zhang,

We’re pleased to inform you that your manuscript has been judged scientifically suitable for publication and will be formally accepted for publication once it meets all outstanding technical requirements.

Kind regards,

Zhihong (Arry) Yao, Ph.D.

Academic Editor

PLOS ONE

Additional Editor Comments (optional):

Reviewers' comments:

Reviewer's Responses to Questions

**Comments to the Author**

Reviewer #1: All comments have been addressed

2. Is the manuscript technically sound, and do the data support the conclusions?

Reviewer #1: Yes

3. Has the statistical analysis been performed appropriately and rigorously?

Reviewer #1: Yes

4. Have the authors made all data underlying the findings in their manuscript fully available?

Reviewer #1: Yes

5. Is the manuscript presented in an intelligible fashion and written in standard English?

Reviewer #1: Yes

Reviewer #1: Thanks for your revision. All the comments have been addressed. The manuscript can be addressed.

**Do you want your identity to be public for this peer review?** For information about this choice, including consent withdrawal, please see our Privacy Policy

Reviewer #1: No

---

## [Editor Report · Acceptance letter]

PONE-D-24-43241R1

PLOS ONE

Dear Dr. Zhang,

I'm pleased to inform you that your manuscript has been deemed suitable for publication in PLOS ONE. Congratulations! Your manuscript is now being handed over to our production team.

Kind regards,

on behalf of

Dr. Zhihong (Arry) Yao

Academic Editor

PLOS ONE